# Predicting trends in atmospheric $CO_2$ across the Mid-Pleistocene Transition using existing climate archives

Jordan R.W. Martin[1], Joel Pedro[2,3], Tessa R. Vance[3]

[1]Institute for Marine and Antarctic Studies, University of Tasmania, Hobart, 7004, Australia

[2]Australian Antarctic Division, Kingston, 7050, Australia

[3]Australian Antarctic Program Partnership, Institute for Marine and Antarctic Studies, University of Tasmania, Hobart, 7004, Australia

*Correspondence to*: Jordan R.W. Martin (jrmartin@utas.edu.au)

**Abstract**

During the Mid-Pleistocene Transition (MPT), ca. 1200–800 thousand years ago (kya), the Earth's glacial cycles changed from 41 kyr to 100 kyr periodicity. The emergence of this longer ice-age periodicity was accompanied by higher global ice volume in glacial periods and lower global ice volume in interglacial periods. Since there is no known change in external orbital forcing across the MPT, it is generally agreed that the cause of this transition is internal to the earth system. Resolving the climate, carbon cycle and cryosphere processes responsible for the MPT remains a major challenge in earth and palaeoclimate science. To address this challenge, the international ice core community has prioritised recovery of an ice core record spanning the MPT interval.

Here we present results from a simple generalised least squares (GLS) model that predicts atmospheric $CO_2$ out to 1.8 Myr. Our prediction utilises existing records of atmospheric carbon dioxide ($CO_2$) from Antarctic ice cores spanning the past 800 kyr along with the existing LR04 benthic $\delta^{18}O_{calcite}$ stack (Lisiecki & Raymo, 2005; hereafter 'benthic $\delta^{18}O$ stack') from marine sediment cores. Our predictions assume that the relationship between $CO_2$ and benthic $\delta^{18}O$ over the past 800 thousand years can be extended over the last one and a half million years. The implicit null hypothesis is that there has been no fundamental change in feedbacks between atmospheric $CO_2$ and the climate parameters represented by benthic $\delta^{18}O$, global ice volume and ocean temperature.

We test the GLS-model predicted $CO_2$ concentrations against observed blue ice $CO_2$ concentrations, $\delta^{11}B$-based $CO_2$ reconstructions from marine sediment cores and $\delta^{13}C$ of leaf-wax based $CO_2$ reconstructions (Higgins *et al*., Yan *et al*., 2019 and Yamamoto *et al*., 2022). We show that there is not clear evidence from the existing blue ice or proxy $CO_2$ data to reject our predictions nor our associated null-hypothesis. A definitive test and/or rejection of the null hypothesis may be provided following recovery and analysis of continuous oldest ice core records from Antarctica, which are still several years away. The record presented here should provide a useful comparison for the oldest ice core records and opportunity to provide further constraints on the processes involved in the MPT.

## 1 Introduction

Ice core records from Antarctica provide comprehensive and continuous records of many climate parameters over the last 800 thousand years, e.g. from the Vostok (Petit *et al.,*1999) and European Project for Ice Coring in Antarctica's Dome-C (EDC) ice cores (Jouzel *et al.*, 2007). One of the major challenges in climate science lies beyond the current threshold of the ice core record. The Mid-Pleistocene Transition (MPT) spans from ca. 1200–800 thousand years ago (kya) (Chalk *et al.,* 2017) and is characterised by a change from regularly paced 40 thousand year (kyr) glacial cycles with thinner glacial ice sheets to quasi-periodic 100 kyr glacial cycles in which ice sheets are more persistent and thicker (Clark *et al.,* 2006, Chalk *et al.,* 2017). To resolve the forcings and feedbacks involved in this transition, multiple nations are targeting recovery of continuous ice cores spanning the MPT under the framework of the International Partnerships in Ice Core Science (IPICS) oldest ice core challenge (IPICS, 2020).

The purpose of the current study is to make a simple prediction of atmospheric $CO_2$ across the MPT. Cross-comparison of our and other predicted $CO_2$ records against observed MPT $CO_2$ data will aid in testing competing hypotheses on the cause of the transition, in particular the role of carbon cycle changes.

The MPT occurred in the absence of any changes to orbital insolation forcing; therefore, the mechanisms behind the MPT must be internal to the earth system (Raymo, 1997; Ruddiman *et al.,* 1989). Multiple hypotheses have been put forward to explain the transition. A common element in many of these is internal climate/earth system changes which allow for the development of thicker, more extensive ice sheets that could endure insolation peaks corresponding to the 23 kyr precession and 41 kyr obliquity cycles, i.e., an increase in the threshold for deglaciation and altered sensitivity to orbital forcings (McClymont *et al.,* 2013; Tzedakis *et al.*, 2017). Indeed, the skipped obliquity cycle hypothesis, proposes that 100 kyr signal seen in spectral analysis of the post-MPT benthic $\delta^{18}O$ stack (e.g. Fig 1A) may be comprised of alternating 80 and 120-kyr signals, i.e. in which the intervening obliquity cycles are skipped. Among the prominent hypotheses to explain an increased threshold for deglaciation are the following three.

1) A long- term decrease in radiative forcing due to a secular reduction in atmospheric $CO_2$ across the transition (e.g. Berger *et al.,* Hönisch *et al.,* 2009; 1999, Raymo *et al.,* 1988). According to this view, reduced radiative forcing drives the formation of larger and more stable ice sheets.

2) Progressive removal of sub-glacial regolith during the 41 kyr glacial cycles. Clark & Pollard (1998) proposed that ice sheet basal sliding prior to the MPT was enhanced by the presence of a low-friction sedimentary regolith layer between the Laurentide ice sheet and the crystalline bedrock. According to this view, progressive removal of this sedimentary layer then favoured the development of larger and more persistent post-MPT ice sheets.

3) Phase-locking of the Northern and Southern Hemisphere ice sheets. In frequency spectra of the global marine benthic $\delta^{18}O$ record (Fig. 1) there is no evidence of the precession (23 kyr) component of northern hemisphere insolation prior to the MPT; the spectra is dominated by the obliquity (41 kyr) component (Fig. 1C). Emergence of significant precession and 100 kyr signals occurs across the MPT (Fig. 1B), and all three components are clearly present after the MPT (Fig. 1A). Raymo *et al.* (2006) suggested that precession-paced changes in northern and southern hemisphere ice volumes may have

occurred prior to the MPT, but are cancelled due to out-of-phase ice volume changes between the two
hemispheres. According to this view, during the MPT the precession-paced changes fall into phase
between the two hemispheres, such that the precession signal emerges (Raymo *et al.,* 2006). In this
view the global synchronisation of ice volume drives the formation of larger and more stable ice sheets.

These hypotheses are not mutually exclusive. For a recent review on the cause of the MPT see Berends *et al.*
(2021a).

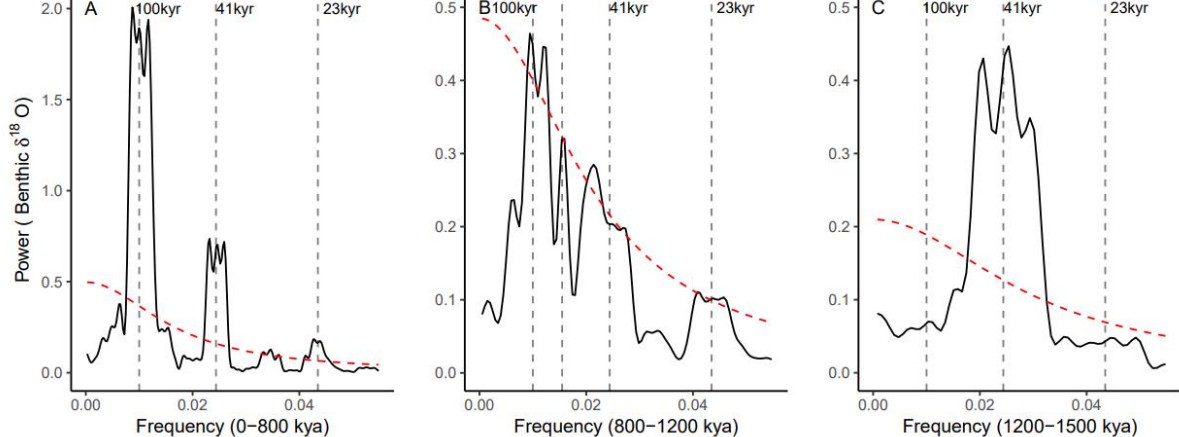



**Figure 1: Thomson Multi–taper Method (MTM) spectral analysis representing relative power of signal periodicity for:**
**A) Benthic δ¹⁸O stack after (0–800 kya) the Mid–Pleistocene Transition (MPT); B) Benthic δ¹⁸O across the MPT (800–**
**1200 kya); C) Benthic δ¹⁸O prior to the onset of the MPT (1200 kya–1500 kya). Each with a robust AR (1) 95 %**
**Confidence interval (red dashed line). Benthic δ¹⁸O stack data from Lisiecki and Raymo (2005).**

For a long-term decrease in radiative forcing by atmospheric $CO_2$ to be the cause of the MPT, the reduction in
$CO_2$ might be expected in both glacial and interglacial stages (Chalk *et al*., 2017). However, low resolution
boron-isotope-based $CO_2$ reconstructions by Hönisch *et al*., (2009), and Chalk *et al*., (2017) suggest that glacial-
stage $CO_2$ drawdown occurred over the MPT in the absence of interglacial $CO_2$ drawdown. Glacial-stage $CO_2$
draw-down across the MPT may be a positive climate–carbon cycle feedback to changes in ice sheet dynamics,
including $CO_2$ drawdown by enhanced iron fertilisation of the Southern Ocean in response to exposed
continental shelves due to lower sea level, as well as planetary drying associated with colder climate conditions
(Chalk *et al*., 2017). Colder glacial temperatures that enhance the solubility of $CO_2$ in the oceans, and reduced
abyssal ocean ventilation has also been implicated in enhanced glacial-stage ocean storage of $CO_2$ (McClymont
*et al*., 2013; Hasenfratz *et al*., 2019).

Testing of hypotheses on the cause of the MPT is currently limited by the lack of a continuous ice core that
spans its duration. The International Partnership in Ice Core Sciences (IPICS) has nominated recovery of such a
record as a key priority in ice core research (IPICS, 2020). Multiple national and international projects have
commenced, or are soon to commence, drilling for 'oldest ice' (see e.g. Shugi, 2022). In this project, we take
inspiration from the "EPICA Challenge" in which the paleoclimate and modeling community was challenged to
predict the global atmospheric carbon dioxide and methane concentrations from 800–400 kya based on the
existing 400 kyr Vostok ice core record (Wolff *et al.*, 2004). Here, we use a generalised least squares (GLS)
model trained on continuous climate archives to predict a $CO_2$ record out 1.8 Mya. We utilise two primary data
sets for the GLS model: the existing 800 kyr ice core composite record of atmospheric $CO_2$ (Bereiter *et al.,*
2015) and the LR04 benthic stack of 57 globally-distributed records of the $^{18}O$ to $^{16}O$ ratio of fossil benthic
foraminifera calcite (hereafter referred to as the LR04 $\delta^{18}O$ benthic stack). The $\delta^{18}O$ ratios in the LR04 benthic
stack are governed primarily by deep ocean temperature and global ice volume at the time the foraminifera
lived, with higher values indicating both increased ice volume and a colder climate. The relationship between
the ice volume and ocean temperature components contributing to the $\delta^{18}O$ benthic stack are not linear.
Separating the two signals remains challenging and has been attempted elsewhere using a range of approaches
from comparison with paired deep ocean temperature proxies (Elderfield *et al.*, 2012), inverse modelling
(Berends *et al.*, 2021b) and spectral analysis (e.g. Huybers and Wunsch, 2009).

Fig. 2 shows a scatter-plot of the LR04 $\delta^{18}O$ benthic stack versus observed ice core $CO_2$ over the past 800 kyr.
Both data sets are binned to equivalent 3-kyr time steps (Methods). The Pearson's correlation coefficient (r)
between the data sets is -0.82 ($p < 0.05$) indicating that ~68% of the variance in observed $CO_2$ is shared with the
LR04 $\delta^{18}O$ benthic stack. This strong relationship provides an initial rationale for using the LR04 $\delta^{18}O$ benthic
stack as an input parameter to predict $CO_2$ beyond 800 kyr. Mechanistically, multiple processes are expected to
contribute to the shared variance. A first order factor is the dependency of $CO_2$ solubility on ocean temperature
(e.g. Millero, 1995). From the simple solubility perspective, colder climate states with increased ice volume and
colder ocean temperatures will drive increased ocean uptake of $CO_2$ (Berends *et al.,* 2021a). However, the
solubility effect only accounts for a portion of observed glacial $CO_2$ drawdown (Archer *et al.*, 2000). Multiple
additional contributors to the shared variance are proposed in the literature. These include (not exhaustively),
direct radiative forcing of ice volume changes by $CO_2$ (e.g. Shackleton *et al.*, 1985); the impact of ice
volume/sea level changes on atmospheric $CO_2$ via ocean productivity and carbonate chemistry changes (e.g.
Broecker, 1982; Archer *et al.*, 2000; Ushie and Matsumoto, 2012); $CO_2$ drawdown during periods of high ice
volume by increased iron fertilisation (e.g. Röthlisberger *et al.*, 2004; Martinez-Garcia *et al.*, 2014) and
enhanced sea ice extent during periods of high ice volume capping the ventilation of $CO_2$ from the ocean
interior at high latitudes (Stephens and Keeling, 2000).

A quantitative separation and attribution of the processes linking global ice volume, ocean temperature and
atmospheric $CO_2$ on millennial to orbital timescales is not currently available (e.g. Archer *et al.*, 2000; Sigman
*et al.*, 2010; Gottschalk *et al.*, 2019) and will not be attempted here. Rather, we make the simple assumption that
the relationships between the LR04 benthic $\delta^{18}O$ stack and $CO_2$ can be extended beyond 800 kya and use
generalised least squares (GLS) regression modelling between benthic $\delta^{18}O$ and $CO_2$ to make a prediction of
$CO_2$ spanning 800–1500 kya. The deliberately simple implicit assumption, and null hypothesis, is that there is
no change to the feedback processes linking benthic $\delta^{18}O$ and $CO_2$ before and after the MPT.

This approach differs to previous more complex model studies that have attempted to reconstruct $CO_2$ using the
LR04 benthic $\delta^{18}O$ stack as an input variable (van de Wal, 2011; Stap *et al.,* 2016, Berends *et al.*, 2021b). The
latter studies use an inverse forward modelling approach, in which climate and ice sheet models of various
complexities are used to capture physical relations between $CO_2$, global temperature and ice volume. For
example, in Berends et al., 2021b the offset between modelled and observed benthic $\delta^{18}O$ is used to calculate a
value for atmospheric $CO_2$ that is iterated back to the inverse model. The $CO_2$ record which minimises the
difference between the modelled and observed benthic stack is then taken as an estimate of how atmospheric
$CO_2$ may have evolved to force coupled climate, deep ocean temperature and land ice volume changes that
reproduce the observed benthic $\delta^{18}O$ signal. Accuracy of the reconstructions in the inverse modelling approach
depends on the ability of the climate and ice sheet models used to capture the correct climate dynamics across
the MPT. Our GLS method is a simpler statistical approach, designed with the specific null hypothesis in mind,
that does not attempt to simulate the physics linking benthic $\delta^{18}O$ signal, land ice volume, global temperature
and $CO_2$. A range of approaches to reconstructing $CO_2$ have been called for and are of value in the context of
forthcoming continuous ice core records across the MPT from oldest ice projects currently underway in
Antarctica [IPICS 2020].

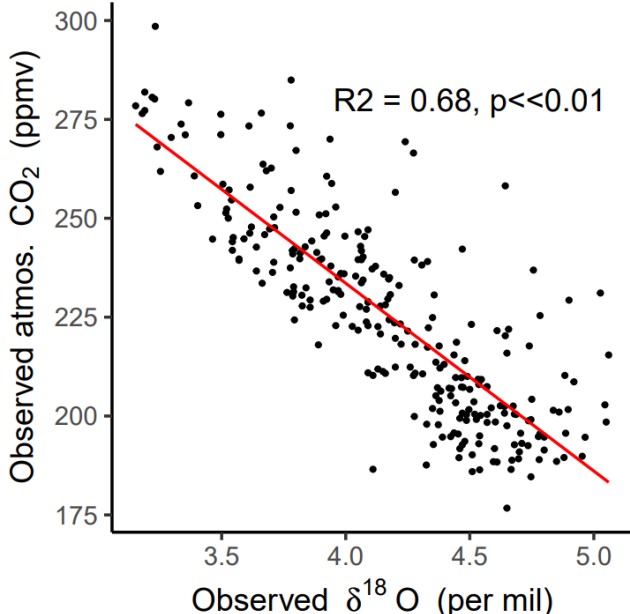

**Figure 2: Scatter plot of the composite observed atmospheric $CO_2$ record (Bereiter *et al.,* 2015) against**
**the LR04 benthic stack of marine $\delta^{18}O$ records (Lisiecki & Raymo, 2005). Red line is a linear line of best**
**fit ($R^2 = 0.68$; $p < 0.05$).**

To test our null hypothesis, in advance of the recovery of a continuous ice core, we compare our predicted $CO_2$
record to two sets of low-resolution ice core data that exist outside the current 800 kyr observed $CO_2$. These data
come from direct $CO_2$ measurements from ancient "blue ice" from the Allan Hills in East Antarctica (hereafter
referred to as BI-$CO_2$) from ca. 1 Mya (Higgins *et al.,* 2015) and 1.5 Mya (Yan *et al.,* 2022). We use the term
blue ice to describe deep, ancient glacial ice that has been brought nearer to the surface of an ice sheet by ice
flow. Blue ice is sampled by cutting trenches or shallow drilling of up to several hundred meters (e.g. Higgins *et*
*al.,* 2015). The vertical migration of blue ice is associated with high deformation making the ice samples
stratigraphically complex and hard to date (Higgins *et al.,* 2015). As a result, blue ice records alone do not
provide a continuous $CO_2$ record across the MPT. In the Discussion, we also compare our predicted record to
existing proxy-$CO_2$ reconstructions from boron-isotope analysis of benthic foraminfera in marine sediment
records (Chalk, *et al*., 2017; Dyez *et al.,* 2018; Guillermic *et al.,* 2022), leaf wax $\delta^{13}C$ carbon isotope ratios
(Yamamoto *et al.,* 2022) and predictions from previous models of various complexities (van de Wal *et al.,* 2011;
Willeit *et al.* 2019; Berends *et al.* 2021b). We conclude with discussion of the implications of our results and
data-comparisons for the understanding MPT dynamics.

**2 Methods**

We use a generalised least squares (GLS) model with an auto-regressive (AR) factor 1 to predict atmospheric $CO_2$
from the LR04 benthic $\delta^{18}O$ stack (Fig. 3A and B). We use GLS because the assumptions of ordinary least squares
(OLS) are violated by the presence of autocorrelation and heteroskedasticity in the regression errors. We selected
the AR(1) correlation factor as it yielded the lowest Akaike information criterion (AIC) value from a test of
multiple correlation factors. The AR(1) process assumes and accounts for dependence of error at a given point in
time on the previous error term. In practise this makes the model assumptions more realistic and improves
parameter estimation where, as in the climate system, observations are dependent on past values.

To obtain common time steps and resolution between the predictor (LR04 benthic $\delta^{18}O$ stack) and response
($CO_2$) variables, we re-grid the LR04 benthic stack and Bereiter *et al.,* (2015) $CO_2$ data into time bins with a
resolution of 3-kyr. The GLS regression model was then applied over the 0 – 800 kyr range of the predictor and
response variables as follows:

$$CO_2 = -33.37 \times \delta^{18}O + 365.15, autoregressive\ (AR) factor: 1$$

Based on the regression model, the $\delta^{18}O$ values of the LR04 Benthic Stack from 800 – 1500 kya were used to
predict $CO_2$ concentration over this range (hereafter referred to as PRED-$CO_2$. To gauge the GLS model
stability we took a bootstrap approach, selecting a random 50% subset of our data (with replacement) and re-
running the model 1000 times to determine 95% confidence intervals for the predictions. While the GLS method
itself addresses autocorrelation, the bootstrap method introduces variability such that each iteration of the model
has different combinations of the original data points (including repeated ones), this variability helps in
assessing the robustness and sensitivity of the model e.g. to variable data and dating uncertainty.

Uncertainties in the independent age scales of both the LR04 stack and the compiled $CO_2$ record are inherited by
our GLS model and its predictions. The LR04 stack includes 57 globally-distributed benthic $\delta^{18}O$ sediment core
records. The age models for these cores are constructed by alignment of their $\delta^{18}O$ signals, followed by tuning
of the stack to a simple ice model based on 21 June insolation at 65°N in a way which maintains relatively
stable global mean sedimentation rates. (Lisiecki & Raymo, 2005). The authors estimate uncertainty of 6 kyr
from 1.5 – 1.0 Mya and 4 kyr from 1 – 0 Mya (Lisiecki & Raymo, 2005). The observed $CO_2$ composite ice core
record for the past 800 kya (Bereiter at al., 2015) uses six independent dating methods for various core locations
both spatially across Antarctica, and stratigraphically for different sections of the same core. The age uncertainty
in the gas timescale has a median over the 0 – 800 kya interval of 2 kyr, but individual uncertainties can reach
up to 5 kyr (Veres *et al* 2013; Bazin *et al*., 2013). The relative age uncertainties between these input variables
may diminish the regression or in some instances lead to spurious correlation. However, we expect any such
effects are minor on the basis that our predictions show little sensitivity to the bootstrap analysis; with a median
$2\sigma$ error of 5.8 ppm from 0 to 1.8 Mya (see Fig. 3B, C and Discussion).

**3 Results**
Fig. 3B shows the time series of our LR04 benthic $\delta^{18}O$ stack-based GLS model predictions of atmospheric $CO_2$
(PRED-$CO_2$) over the past 800 kyr, in comparison to the observed ice core $CO_2$ record from Bereiter at al.,
(2015). The correlation coefficient ($R^2$) between the predicted and observed records is 0.68 (p <<0.01). Our
PRED-$CO_2$ record out to 1.8 Mya with shaded 95% CIs from the bootstrap analysis is also shown, overlain with
observed Allan Hills blue ice $CO_2$ (BI-$CO_2$) datasets of age $1000 \pm 89$ kya (Higgins *et al.,* 2015) and 1.5 Mya $\pm$
213 kyr (Yan *et al.,* 2022).

We evaluate the PRED-$CO_2$ record against the observed $CO_2$ data according to criteria of mean concentrations
across the common intervals, and mean concentrations in the glacial and interglacial subsets of the data. First,
the mean $CO_2$ concentration over the common intervals (Fig 3C). From 0 – 800 kya the mean concentration in
observed (Bereiter at al., 2015) and PRED-$CO_2$ data are in close agreement ($225.2 \pm 3.03$ ppm versus the
predicted $225.2 \pm 2.5$ ppm respectively; uncertainties are 95% confidence intervals, i.e. $1.96\sigma$). In the $1000 \pm 89$
kya interval (i.e. averaged across the age uncertainty of the Higgins *et al.* (2015) blue ice data) the BI-$CO_2$
concentration is ~11 ppm higher than PRED-$CO_2$ ($246.7 \pm 8.4$ ppm versus the predicted $235.3 \pm 3.9$ ppm), this
difference is not significant at the 95% confidence level. For the 1.5 Mya $\pm$ 213 kyr interval, the mean BI-$CO_2$
concentration is ~9 ppm lower than PRED-$CO_2$ ($231.9 \pm 5.6$ ppm versus the predicted $240.7 \pm 2.1$ ppm), which
is marginally significant at the 95% level. Comparisons of mean levels across intervals spanning multiple glacial
and interglacial cycles may be biased if (as is likely) the blue ice data is not sampling glacial and interglacial
values with the same uniformity as a continuous record.

To address this, we define the glacial and interglacial thresholds of PRED-$CO_2$ to be respectively the lower and
upper 25th percentiles of the LR04 $\delta^{18}O$ predictor variable (following Chalk *et al.,* 2017). Filtering the observed
(Bereiter at al., 2015) $CO_2$ record and our predicted $CO_2$ record according to these definitions we find a very
close match for glacial ($202.0 \pm 3.2$ versus the predicted $199.7 \pm 1.6$ ppm) and interglacial intervals ($253.9 \pm 4.1$
ppm versus the predicted $253.1 \pm 2.3$ ppm), over the past 800 kya (see Fig. 3D for these comparisons). For blue
ice (BI-$CO_2$) data, a corresponding LR04 isotope signal could not be confidently applied to the measured $CO_2$
concentration due to the uncertainties associated with blue ice dating; therefore, we defined the glacial and
interglacial thresholds of blue ice data according to the top (interglacial) and bottom (glacial) 25th percentiles of
actual $CO_2$. Applying this to the $1000 \pm 89$ kya interval finds that observed BI-$CO_2$ data is ~9 ppm higher than
PRED-$CO_2$ during the glacial stages ($226.2 \pm 4.0$ ppm versus the predicted $217.6 \pm 2.3$ ppm) and ~15 ppm
higher than PRED-$CO_2$ during the interglacial stages ($271.3 \pm 4.5$ versus the predicted $256.3 \pm 3.8$ ppm). These
differences are significant with respect to the constrained uncertainties. During the 1.5 Mya $\pm$ 213 kyr interval,
the mean BI-$CO_2$ concentration did not show any significant difference to PRED-$CO_2$ in interglacial stages
($254.1 \pm 10.3$ versus the predicted $257.2 \pm 1.7$ ppm. During glacial stages there is a small 2.9 ppm difference

between the upper estimate of BI-CO$_2$ and the lower estimate of PRED-CO$_2$ (218.4 ± 1.3 and 224 ± 1.4 ppm

respectively, see Fig 3D). In our view these results, notwithstanding the 2.9 ppm difference at 1.5 Mya, do not

give sufficient cause to reject the GLS model. Furthermore, the comparison indicates that PRED-CO$_2$ is not

drifting systematically away from the existing observed BI-CO$_2$ data (Fig 3D). The differences could of course

be a failing in the model, potential biases in the blue ice data, dating uncertainty and/or other unconstrained

uncertainties (see Discussion for blue ice caveats).

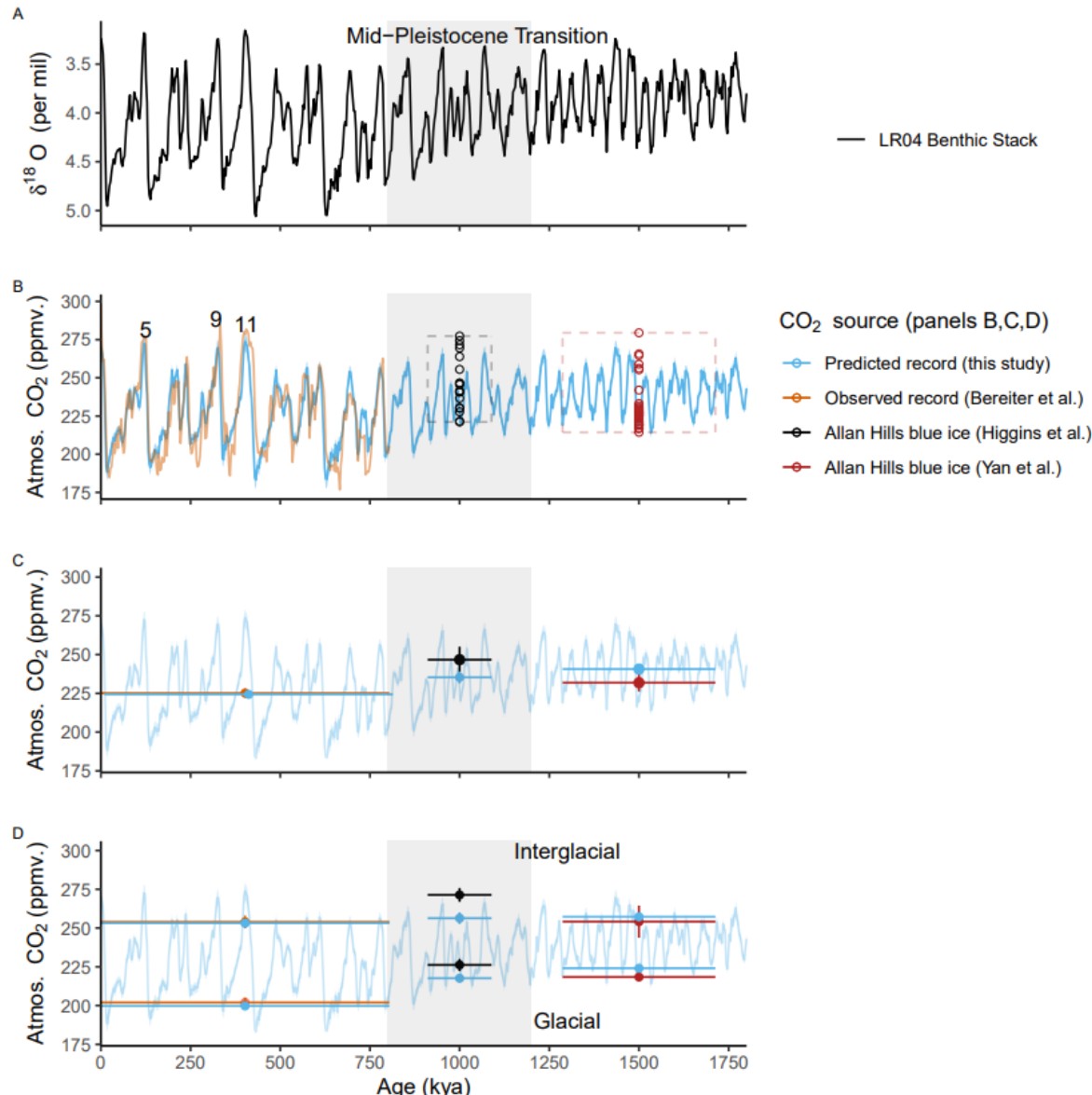

**Figure 3: A) The LR04 Benthic Stack of 57 globally distributed δ$^{18}$O records (Lisiecki & Raymo, 2005).**

**B) Comparison of our PRED-CO$_2$ (ppm) record to the current continuous composite record (0–800 kya);**

**and to direct CO$_2$ measurements from Allan Hills blue ice cores (BI-CO$_2$) ca. 1 Mya (± 89 kyr) (Higgins *et***

***al.,* 2015) and ca. 1.5 Mya (± 213 kyr) (Yan *et al.,* 2022). Age uncertainty boundaries for the BI-CO$_2$ data**

**are represented by dashed box boundaries. Marine isotope stages 5, 9, and 11 are numbered on the plot**

**according to Lisiecki & Raymo (2005). Blue shading around PRED-CO$_2$ is the 95% CI from bootstrap**

**analysis. C) Mean concentrations of the PRED-CO$_2$ and observed composite CO$_2$ records over the range**

**of the observed composite record (offset for clarity), and the mean concentrations of the PRED-CO$_2$ and BI-CO$_2$ data at 1 Mya and again at 1.5 Mya averaged over the age uncertainty range of each BI-CO$_2$ data set. D) As for C) however filtered by the upper and lower 25$^{th}$ and 75$^{th}$ percentiles to estimate glacial and interglacial periods.**

We now consider long-term trends in interglacial and (separately) glacial CO$_2$ levels across the past 1.8 Myr in PRED-CO$_2$ and in the existing ice core CO$_2$ data. For PRED-CO$_2$ there is no significant difference between CO$_2$ concentrations in the interglacial stages of the 1.5 Mya ± 213 kya, 1000 ± 89 kya and 0–800 kya windows (Fig 4 D, blue bars). In the ice core observations, interglacial levels at 1.5 Mya in BI-CO$_2$ are also within the uncertainties of those in the 0–800 kya interval. Notably, the BI-CO$_2$ concentrations in the 1000 ± 89 kya interval appear elevated with respect to the 0–800 kyr and 1.5 Mya ± 213 kya intervals, however this elevated (ca. 271 ppm) level is consistent with the observed interglacial CO$_2$ concentration during interglacials 5, 9 and 11 (Fig 3B). Overall, there is no indication in the observed ice core CO$_2$ data or in PRED-CO$_2$ for a long-term trend in *interglacial* CO$_2$ levels across the past 1.8 Myr.

In comparison, there are significant declines in glacial CO$_2$ levels across the MPT in PRED-CO$_2$ and the observed ice core data. For PRED-CO$_2$, glacial CO$_2$ concentrations are not significantly different during the 1.5 Mya ± 213 kya and 1000 ± 89 kya windows. However, across the MPT, PRED-CO$_2$ glacial concentrations drop by ~18 ppm (Fig 3D). This pattern is similar to the observed BI-CO$_2$ data, where glacial CO$_2$ levels show no decline between the 1.5 Mya ± 213 kya and 1000 ± 89 kya windows (indeed there is a marginal increase from 218.4 ± 1.3 to 226.2 ± 4.0 ppm, respectively), before falling by 24 ppm to the 0–800 kyr observed glacial mean of 202.0 ± 3.2 ppm (Fig 3D). Glacial-stage draw-down of CO$_2$ across the MPT in the absence of interglacial draw-down is consistent with previous observations based on the boron-isotope-based CO$_2$ reconstructions (e.g., Chalk *et al*., 2017; Hönisch *et al*., 2009 and see Discussion). In the following section we also compare PRED-CO$_2$ data to boron-isotope-based and other CO$_2$ proxy records covering the 0 to 1.8 Myr interval.

**4 Discussion**

Our objective with this manuscript was to generate the simplest reasonable model to predict CO$_2$ from the LR04 $\delta^{18}$O benthic stack and to test the predictions against available observations. It is possible that the fit between observed and our predicted CO$_2$ data could be further improved using a non-linear approach. However, we refrain from a non-linear approach for several key reasons. First, a scatter plot of the LR04 $\delta^{18}$O benthic stack versus observed ice core CO$_2$ over the past 800 kyr yields a Pearson's correlation coefficient (R) of -0.82 (Fig. 2), indicating that ~68% of the variance in observed CO$_2$ is shared with the benthic stack. This is similar to that reported in ordinary linear least-squares regression (R$^2$=0.70) by Berends *et al.* (2021b). Importantly, there is no evidence in this scatter plot for departure from the linear relationship at high or low CO$_2$ or benthic $\delta^{18}$O levels. Second, following the approach of Chalk *et al*., 2017 and interpreting the upper 25$^{th}$ percentile of CO$_2$ data as representing mean interglacial stage CO$_2$ and the lower 25$^{th}$ percentile of CO$_2$ data as representing mean glacial stages CO$_2$ levels, we see that our predicted interglacial mean value for the past 800 kyr (253.1 ± 2.3 ppm) closely overlaps with the observed interglacial mean value (253.9 ± 4.1 ppm) and similarly, the predicted glacial stage mean (199.7 ± 1.7 ppm) closely overlaps with the observed glacial stage mean (202.0 ± 3.2 ppm). Third,

the predictions are remarkably insensitive to bootstrap analysis in which 50 % of that data are omitted with each
iteration of the GLS model. Such insensitivity to the bootstrap analysis and accurate prediction of glacial and
interglacial state $CO_2$ values would be unlikely in the case of major non-linear dependencies between the LR04
predictor and $CO_2$ response variables. Fourth, non-linear approaches would risk generating an improved fit due
to statistical artefacts that do not meaningfully relate to any dependence between benthic $\delta^{18}O$ and $CO_2$. Finally,
the specific causes and sources and sinks involved in glacial to interglacial and millennial-scale $CO_2$ variations
remain poorly constrained (e.g. Archer *et al.*, 2000; Sigman *et al.*, 2010; Gottschalk *et al.*, 2019). Given this
process-uncertainty, the GLS model fits our criteria of the simplest reasonable model. Further, the use of benthic
$\delta^{18}O$ to predict atmospheric $CO_2$ has precedence; in response to the EPICA challenge (Wolff et al., 2004) N.
Shackleton predicted atmospheric $CO_2$ out to 800 kyr, based on a number of benthic $\delta^{18}O$ records from the East
Pacific (Wolff, 2005).

There are several caveats with blue ice data that may affect its use to evaluate our GLS model predictions. The
blue ice data may have been subject to diffusional smoothing of $CO_2$ (e.g. Yan *et al.,* 2019), which would act in
the direction of elevating the (lower 25[th] percentile) assumed glacial concentrations above the glacial
atmospheric values and reducing the (upper 25[th] percentile) assumed interglacial concentrations. There is also
the potential for artificially elevated $CO_2$ concentrations in blue ice due in-situ respiration of $CO_2$ due to
microbial activity in detrital matter. Respiration effects are screened for by measurements of $\delta^{13}C$ of $CO_2$,
however it is difficult to demonstrate that all samples are unaffected (Yan *et al*., 2019). These uncertainties
support our argument that the GLS-model predictions are not rejected by the available observed BI-$CO_2$ data.

We consider the BI-$CO_2$ data to provide the most reliable measurements of $CO_2$ concentration, in the absence of
a continuous ice core record across the MPT. However, further comparison of our $CO_2$ predictions can also be
made against $CO_2$ proxy data from non-ice core archives (Fig 4A). We consider here $\delta^{11}B$-based atmospheric
$CO_2$ reconstructions (Chalk *et al.,* 2017, Dyez *et al.* 2018 and Guillermic *et al.* 2022) and a recent atmospheric
$CO_2$ reconstruction from $\delta^{13}C$ of leaf wax (Yamamoto *et al.,* 2022). The continuous $\delta^{11}B$-based reconstructions
of Dyez *et al.*, (2018) overlap PRED-$CO_2$ from ~1.38 – 1.5 Mya while the Chalk *et al.*, (2017) reconstruction
overlaps PRED-$CO_2$ from 1.09 – 1.43 Mya. Discrete reconstructions from Guillermic *et al.* (2022) are
distributed non-uniformly across the ~800 to 1.5 Mya interval. For the two continuous $\delta^{11}B$-based
reconstructions (Chalk *et al.*, (2017) and Dyez *et al.*, (2018)) the glacial $CO_2$ levels appear consistent with the
PRED-$CO_2$ record, within their reported 30 – 60 ppm uncertainties. However, $\delta^{11}B$-based interglacial stages in
these reconstructions exceed those of the PRED-$CO_2$ record (Fig. 4A). The Guillermic *et al.* (2022)
reconstructions suggest a larger range of $CO_2$ concentrations than the overlapping intervals of PRED-$CO_2$ and of
the two continuous $\delta^{11}B$-based reconstructions (Fig. 4A). The large range of the Guillermic *et al.* (2022) data
and the high interglacial maxima in the Chalk *et al* (2017) and Dyez *et al.*, (2018) data, all significantly exceed
the range and interglacial maxima from the BI-$CO_2$ estimates. These discrepancies internally between different
$\delta^{11}B$-based $CO_2$ reconstructions and between the $\delta^{11}B$-based reconstructions and the BI-$CO_2$ data, may be due to
uncertainties associated with the $\delta^{11}B$ proxy transfer function. The $\delta^{11}B$-based $CO_2$ reconstructions are
dependent on assumptions about multiple components of the carbonate system, including local marine carbon
chemistry and the $CO_2$ saturation state in the past (Hönisch *et al*., 2009). Evidence that $\delta^{11}B$-based
reconstructions may overestimate interglacial stage $CO_2$ is also seen in data from Chalk *et al.*, (2017) spanning
ca. 0–250 kya, where the $\delta^{11}B$-based interglacial $CO_2$ levels exceed the continuous ice core $CO_2$ record by up to
ca. 30 ppm.

By comparison, the $\delta^{13}C$ of leaf wax data (Yamamoto *et al.,* 2022) has a similar glacial to interglacial range as
PRED-$CO_2$, but a ca. 20ppm lower mean concentration than our predictions (Fig 4A). Hence, our PRED-$CO_2$
data fall lower than interglacial $\delta^{11}B$-based interglacial levels but are higher than the $\delta^{13}C$ of leaf-wax based
estimate. The strong spread between these different proxies and the large associated uncertainty of the
alternative marine and leaf wax proxy-$CO_2$ reconstructions mean that we do not find cause from the existing
$CO_2$ proxy data to reject our predictions nor our associated null-hypothesis.

We also compare our predictions to existing more complex model simulations (Fig 4B.). First, against a
transient simulation using an intermediate-complexity earth system model (CLIMBER-2) by Willeit *et al.*
(2019). This study suggests a combination of gradual regolith removal and atmospheric $CO_2$ decline can explain
the long-term climate variability over the past 3 Myr. Second, against a longer-term reconstruction by van de
Wal *et al.* (2011), which uses benthic $\delta^{18}O$ that utilises deep-sea benthic isotope records to reconstruct a
continuous $CO_2$ record over the past 20 Myr. Third, a $CO_2$ reconstruction based on an inverse forward-
modelling approach forced by the LR04 benthic stack, in which the forward model is incrementally updated
through interaction with general circulation model snapshots and the ANICE 3-D ice-sheet-shelf model
(Berends et al. 2021b). Our simple GLS model demonstrates a similar long-term trend and timing of glacial-
interglacial signals and an atmospheric $CO_2$ level that sits approximately mid-way between the van de Wal *et al.*
(2011), and Willeit *et al.* (2019) models and is remarkably similar to the Berends *et al.* (2021b) reconstruction,
despite their different approach. Notably the Berends et al. reconstruction shows greater glacial to interglacial
amplitude in the $CO_2$ signal compared to our GLS-model. The decreasing linear trend in $CO_2$ in Willeit et al.
(2019), which is not seen in the other reconstructions, was directly prescribed in that study to induce Northern
Hemisphere glaciation at 2.6 Myr ago.

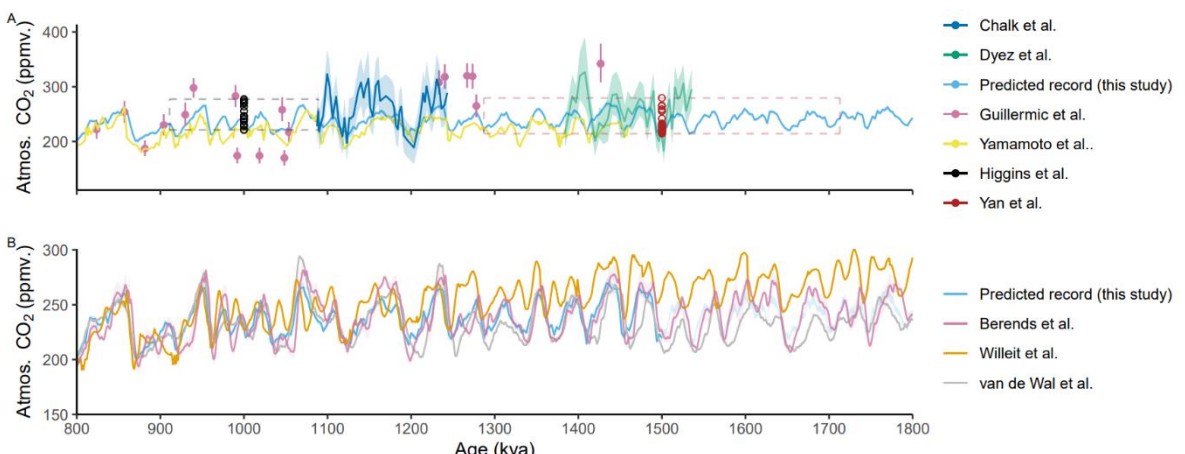


**Figure 4: A) Predicted $CO_2$ (this work) compared to observed, proxy $CO_2$ estimates from a range of other**
**sources: $\delta^{11}B$-based pCO₂ reconstructions and measurements by Dyez *et al.* (2018), Guillermic *et al.***
**(2022); Chalk *et al.*, (2017); blue ice $CO_2$ measurements by Yan *et al.* (2019) and Higgins *et al.* (2015);**
**δ¹³C leaf wax proxy reconstructions by Yamamoto *et al.* (2022). The dashed boxes indicate the dating**
**uncertainty and range of the respective BI-CO₂ records. B) Our predicted record compared to various**
**model simulations: a regolith removal hypothesis simulation by Willeit *et al.* (2019); and inverse-model**
**based CO₂ reconstructions by van de Wal *et al.* (2011), and Berends *et al.*, (2021b).**

A complete and critical test of our and other $CO_2$ predictions awaits the upcoming analysis of the continuous
oldest ice core records. We now discuss some potential applications of the PRED-CO₂ record for hypothesis
testing on the cause of the MPT.

PRED-CO₂ shows a long-term decline in glacial $CO_2$ across the MPT, but no long-term decrease in interglacial
$CO_2$. This pattern is consistent with the boron-isotope-based $CO_2$ reconstructions shown earlier, where it is often
described as an increase in the interglacial to glacial $CO_2$ difference (e.g., Chalk *et al.*, 2017; Hönisch *et al.*,
2009). Chalk *et al*, (2017) concludes that the MPT was initiated by a change in ice sheet dynamics and that
longer and higher-ice volume post-MPT ice ages are sustained by carbon cycle feedbacks, in particular dust
fertilisation of the Southern Ocean. The fact that our LR04-based prediction of $CO_2$ captures this same trend,
with predicted glacial $CO_2$ fairly constant from 1.5 to ca. 1.0 Mya before declining from 1.0 to 0.6 Mya, reflects
that the LR04 benthic stack also features an increase in the interglacial to glacial benthic $\delta^{18}O$ difference across
this same interval, which is dominated by the glacial stage changes (Fig 3A.). Here, a comparison of PRED-CO₂
to a realised continuous oldest ice core record will be of value. The agreement or disagreement would inform on
the proportionality of the $CO_2$ coupling with ice volume; if there were a major new or non-linear process across
the MPT that changed the nature of coupling between $CO_2$ and ice volume the PRED-CO₂ and observed $CO_2$
records would be expected to diverge.

Another avenue to use the PRED-CO₂ record for hypothesis testing on the cause of the MPT concerns the phase
locking hypothesis. The phase locking hypothesis is proposed to explain the absence of precession-related (23
kyr) periods in the LR04 benthic stack prior to the MPT (Fig 1), despite the strong precession cycle in insolation
(Raymo *et al.*, 2006, Morée *et al.*, 2021). The key concept is that prior to the MPT the Northern Hemisphere and
Antarctic ice sheets were responsive (in ice volume) to insolation changes in the precession band, but because
precession forcing is out of phase between the hemispheres, the ice volume changes were opposing between the
hemispheres and therefore cancelled in the benthic stack. This cancellation of the precession signal left
insolation forcing in the 41 kyr obliquity band to dominate globally integrated ice volume changes expressed in
the benthic stack. A transition from a smaller and more dynamic terrestrial-terminating Antarctic ice sheet to a
larger and more stable marine-terminating ice sheet with cooling climate across the MPT (e.g. Elderfield *et al.*,
2012) is then proposed to remove sensitivity of Antarctic ice volume to local precession forcing in favour of
quasi-100 kyr ice volume changes that are in phase between the hemispheres (Raymo et al., 2006).

Recently presented data from Yan *et al.* (2022), lend some support to the phase locking hypothesis, specifically
with evidence that pre-MPT Antarctic temperature (and by extension ice volume) is positively correlated with a
local precession-band insolation proxy based on the oxygen to nitrogen ratio of trapped air (Yan *et al.*, 2022).
Whereas the correlation becomes negative in the blue ice and continuous ice core data in the post-MPT record.
If Yan *et al*., (2022) is correct and the phase locking hypothesis holds, then an implication is that prior to the
MPT, Antarctic climate, Antarctic ice volume and by extension Southern Ocean climate conditions, would fall
out of phase with the LR04 benthic stack. To now extend the argument to potential impacts on $CO_2$ exchange, if
the phase locking hypothesis holds, then prior to the MPT the Antarctic and Southern Ocean climate conditions
and by extension the Southern Ocean mechanisms of $CO_2$ exchange described earlier, would also be expected to
fall out of phase with the benthic stack. Since our regression model assumes continuation of the in-phase
relationship between the benthic stack and Antarctic and Southern Ocean climate conditions (as inherited from
the post-MPT training data) we would expect to see major disagreement between our pre-MPT $CO_2$ predictions
and a realised oldest ice continuous ice core $CO_2$ record.

**5 Summary and Conclusions**
In this study we have used a simple generalised least squares (GLS) model to predict atmospheric $CO_2$ from the
LR04 benthic $\delta^{18}O$ stack for the period spanning the mid-Pleistocene transition, 800–1800 kyr. Our $CO_2$
prediction is therefore based on the assumption that the physical processes linking $CO_2$, sea level, global ice
volume and ocean temperature over the past 800 kyr do not fundamentally change across the 800–1800 kya time
period. The null-hypothesis is deliberately simplistic on the basis that differences between our predictions and
observed or proxy $CO_2$ records may be revealing of the physical processes involved in the mid-Pleistocene
Transition.

We made initial tests of the null hypothesis by comparing our predicted $CO_2$ record to existing discrete blue ice
$CO_2$ records and other non-ice-core proxy-$CO_2$ records from the 800–1800 kyr interval. Our predicted $CO_2$
concentrations do not show any systematic departure from observed blue ice $CO_2$ concentrations. The
predictions are marginally lower (during glacial *and* interglacial stages) than those observed in blue ice from
$1000 \pm 89$ kya and marginally higher than observed in blue ice data from 1.5 Mya $\pm$ 213 kyr. Our predictions
were generally lower than interglacial $\delta^{11}B$-based-$CO_2$ reconstructions, but higher than recent $\delta^{13}C$ of leaf-wax
based $CO_2$ reconstructions. Overall, we do not find clear evidence from the existing blue ice or proxy $CO_2$ data
to reject our predictions nor our associated null-hypothesis. The definitive test of our and other $CO_2$ predictions
therefore awaits the future analysis of the upcoming continuous oldest ice core records. The PRED-$CO_2$ record
presented here should provide a useful comparison to forthcoming oldest ice core records and opportunity to
provide further constraints on the processes involved in the MPT.

**Author contributions**
Project design by JBP, TRV and JRWM and supervision by TRV and JBP. Data analysis and figures by JRWM
with input from all authors. Writing led by JRMV and JBP. All authors contributed to and agreed on the final
version of the manuscript.

**Competing interests**
The authors declare that they have no competing interests.

**Disclaimer**

This study, to the best of the author(s) knowledge and belief, contains no material previously published or

written by another person, except where due reference is made in the text of the study.

**Acknowledgements**

We acknowledge assistance from Simon Wotherspoon (Institute for Marine and Antarctic Studies) in

appropriate model selection methods. We thank Lorraine Lisiecki and Constantijn Berends's for their

constructive reviews, which greatly improved the manuscript. This research was supported by the Australian

Government through Australian Antarctic Science projects 4632, the Million Year Ice Core (MYIC) Project and

by the Australian Government Department of Industry Science Energy and Resources, grant ASCI000002.

**Data availability**

The PRED-$CO_2$ data presented here will be publicly archived at the Australian Antarctic Data Centre

(https://data.aad.gov.au/metadata/AAS_4632_Martin_etal_CP_2024

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
