# Peer review of "Predicting trends in atmospheric $CO_2$ across the Mid-Pleistocene Transition using existing climate archives"

_EGUsphere, 2022_

## Author Comment (AC7)

An earlier incomplete version of the review response was uploaded on 24/11. Please refer to the final response below. Reviewer comments are in black text and our response in blue. We do not include a tracked changes version of the manuscript or a revised manuscript at this point as the editor request is to respond to the comments only and not to prepare a revised manuscript. We do provide in the response below edits that we would include in a revised manuscript.

**RC1**

This manuscript presents a null hypothesis prediction for $CO_2$ across the MPT based on generalized least squares regression between Late Pleistocene $CO_2$ records from Antarctic ice and the LR04 global benthic $\delta^{18}O$ stack, a proxy for changes in global ice volume and deep water temperature. The regression-based predictions are then compared with sparse MPT $CO_2$ estimates from blue ice (Higgins et al, 2015) and boron isotopes (Chalk et al, 2017) with respect to mean value and glacial-interglacial range and compared to trend in $CO_2$ from an intermediate complexity model run across the MPT (Willeit et al, 2019). The authors argue that misfit between pre-MPT $CO_2$ values and the regression-based predictions would be evidence for a change in climate-carbon cycle-cryosphere dynamics across the MPT.

*R1 Comment: The analysis appears to be performed well, and I have only a few concerns about the interpretation of the results. However, my main concern is that the work is too simple. I would encourage the authors to add more intellectual substance to the paper by exploring perhaps nonlinear regression between benthic $\delta^{18}O$ and $CO_2$…*

**Response:** We appreciate the positive comments about the analysis and acknowledge the simplicity of the generalised least squares (GLS) model. In our view the simplicity of the model is a strength – our objective with this manuscript was to generate the simplest reasonable model to predict $CO_2$ from the $\delta^{18}O$ benthic stack and to test the predictions against available observations. In response to this review and comments from other reviews we have substantially expanded the work in three main areas to add more substance:

1. We bring in substantial additional data to evaluate our predicted record $CO_2$ record, specifically: Allan Hills Blue Ice $CO_2$ data from Yan *et al.,* (2019) at 1.5 Mya; $CO_2$ proxy reconstructions from leaf wax from Yamamoto *et al.,* (2022); δ11B reconstructions by Dyez *et al.,* (2018), and Guillermic *et al.,* (2022) and a high resolution $CO_2$ reconstruction by van de Wal *et al.* (2011).
2. We add discussion of underlying mechanisms relating benthic $\delta^{18}O$ and $CO_2$.
3. In light of the additional comparisons with blue and proxy $CO_2$ records from (1.) we find that our regression-based prediction of pre-MPT $CO_2$ is not rejected by the existing data and we discuss the implications.

More on each of these points is provided in the responses to follow. Regarding the specific point about exploring non-linear regression plan to add new text to the manuscript to justify the GLS approach as follows:

> "Our objective with this manuscript was to generate the simplest reasonable model to predict $CO_2$ from the $\delta^{18}O$ benthic stack and to test the predictions against available observations. It is possible that the fit could be further improved using a non-linear approach, however we deliberately refrain from that for several key reasons. First, a scatter plot of the LR04 $\delta^{18}O$ benthic stack versus observed ice core $CO_2$ over the past 800 kyr yields a Pearson's correlation coefficient (r) of -0.82 (see Figure R1-1), indicating that ~68% of the variance in observed $CO_2$ is shared with the benthic stack. There is no evidence in this

scatter plot for departure from the linear relationship at high or low $CO_2$ or benthic $\delta^{18}O$ levels. Second, following the approach of Chalk et al., 2017 and interpreting the upper 25[th] percentile of $CO_2$ data as representing mean interglacial stage $CO_2$ and the lower 25[th] percentile of $CO_2$ data as representing mean glacial stages $CO_2$ levels, we see that our predicted interglacial mean value for the past 800 kyr (253.1 ± 2.3 ppm) closely overlaps with the observed interglacial mean value (253.9 ± 4.1 ppm) and similarly, the predicted glacial stage mean (199.7 ± 1.7 ppm) closely overlaps with the observed (202.0 ± 3.2 ppm). Furthermore, the predictions are remarkably insensitive to bootstrap analysis in which 50 % of that data are omitted with each iteration of the GLS model (Fig 1). Such insensitivity to the bootstrap analysis and accurate prediction of glacial *and* interglacial state $CO_2$ values would be unlikely in the case of major non-linear dependencies between the LR04 predictor and $CO_2$ response variables. Third, non-linear approaches would risk generating an improved fit due to statistical artefacts that do not meaningfully relate to any dependence between benthic $\delta^{18}O$ and $CO_2$. Finally, the specific causes and sources and sinks involved in glacial to interglacial and millennial-scale $CO_2$ variations still remain poorly constrained (e.g. Archer et al., 2000; Sigman et al., 2010; Gottschalk et al., 2019), and given that process-uncertainty, the GLS model fits our criteria of the simplest reasonable model."

[Figure]

*Figure R1-1: Observed continuous ice core $CO_2$ versus the LR04 benthic $\delta^{18}O$ stack over the past 800,000 years. Pearson's correlation coefficient r = -0.82. $CO_2$ data from Bereiter et al., (2015), LR04 benthic stack from Lisiecki & Raymo, (2005).*

[Figure]

*Figure 1 (from manuscript): A) Comparison of our PRED-CO$_2$ (ppm) record to the current continuous composite record; CO$_2$ estimates from boron isotope analysis of benthic foraminifera shells (BOR-CO$_2$) (Chalk, et al., 2017), and direct CO$_2$ measurements from Allan Hills blue ice core data (BI-CO$_2$) (Higgins et al., 2015). Indicators for age uncertainty boundaries (± 89 ky) of the blue ice represented by dashed boundaries. Blue shading around the PRED-CO2 curve represents 95% confidence intervals generated from the bootstrap analysis (Methods).*

**R1 Comment:** *"…discussing in more depth the underlying mechanisms relating benthic δ$^{18}$O and CO$_2$ to say more about the implications of potential misfit between CO$_2$ and the regression-based estimate."*

This is a good suggestion and in response we would revise the manuscript (nominally around lines 58-59) to give more detail on potential underlying mechanisms (while also acknowledging that there is no consensus on these):

"The δ$^{18}$O of fossil benthic foraminifera calcite is governed by ocean temperature and global ice volume at the time the foraminifera lived, with higher values indicating both increased ice sheet volume and a colder climate. The first order rationale in using the LR04 δ$^{18}$O benthic stack as an input parameter to predict CO$_2$ is based on the known relationship of ocean temperature with CO$_2$ solubility (e.g. Millero, 1995). The solubility of CO$_2$ in the ocean increases with falling sea surface temperatures, particularly in high-latitude deep-water formation regions, where colder ocean temperature drive increased uptake of CO$_2$ by the ocean, reducing the atmospheric CO$_2$ concentration (Berends et al., 2021). However, we note that the magnitude of glacial cooling can only account for a portion of observed glacial CO$_2$ drawdown (Archer et al., 2000) and multiple other dependencies between CO$_2$ and ocean temperature and/or ice volume are also likely at play in explaining the observed shared variance. These may include (not exhaustively), direct radiative forcing of ice volume changes by CO$_2$ (e.g. Shackleton et al., 1985), second order effects on atmospheric CO$_2$ from changing ice volume, including the impact of ice volume/sea level on atmospheric CO$_2$ via ocean productivity and carbonate chemistry changes (e.g. Broecker et al., 1982; Archer et al., 2004; Ushie and Matsumoto 2012), CO$_2$ drawdown during periods of high ice volume by increased iron fertilization (e.g. Rothlisberger et al., 2004; Martinez-Garcia et al., 2014) and enhanced sea ice extent during periods of high ice volume capping the ventilation of CO$_2$ from the ocean interior at high latitudes (Stephens and Keeling, 2000)."

**R1 Additional Specific Concerns**

*R1 Comment: Abstract, line 18: I think the authors meant benthic foraminiferal stable isotope (δ18O). The δ18O data used is from foraminiferal calcite, not "water."*

Thank you. Corrected to "the existing benthic foraminferal calcite $\delta^{18}O$ record from marine sediment cores."

*R1 Comment: Line 118: It is not clear what the authors mean by "This trend is seen in our predicted record, and in the filtered BI-CO₂ and BOR-CO₂ data (Fig. 1C)." The previous sentence describes glacial stage CO₂ draw-down and the absence of an interglacial draw-down. In Fig. 1C, it appears that this description holds for the predicted CO₂ record (i.e., glacial draw-down but steady interglacial values). However, the BI-CO₂ and BOR-CO₂ data show a change in BOTH glacial and interglacial CO₂ compared to the post-MPT average. The text should be revised to make clear which trends are similar between the predictions and observations and which are different.*

We take the opportunity to clarify and expand on this section. We would include a comparison to the Yan et al., 2019 BI-CO₂ data from 1.5 Mya and draw back on quantitative comparison to BOR-CO₂ data on account of its larger uncertainties than BI-CO₂ data. Nominal revised text around Line 118 as follows:

> "Previous studies conclude that glacial stage draw-down of CO₂ occurred across the MPT in the absence of interglacial draw-down; i.e. glacial stage atmospheric CO₂ concentrations decline with time across the MPT, whereas interglacial stage CO₂ concentrations remain comparatively stable (e.g., Chalk et al., 2017; Hönisch et al., 2009). This trend is also seen in our predicted record (Fig 1A & 1C). For example, our predicted glacial CO₂ concentration ca. 1Myr is 217.6 ± 2.3 ppm (central lower blue bar Fig 1C), which is significantly higher than the predicted mean of 199.7 ± 1.7 ppm for glacial stages of the past 800 kyr (lower left blue bar Fig 1C). Comparing this to the observed Higgins et al., (2015) BI-CO₂ data (and hereafter interpreting the lowest 25ᵗʰ percentile of the BI-CO₂ as representing glacial stage atmospheric CO₂, after Chalk et al., (2017)), the BI-CO₂ data suggest a concentration of 226.2 ± 4.0 ppm for glacial stages ca. 1Myr (lower black bar Fig 1C), which is significantly higher than the mean observed glacial atmospheric CO₂ concentration over the past 800 kyr of 202.0 ± 3.2 ppm (left orange bar Fig 1C). Turning to the interglacials, our GLS model predicts CO₂ concentration during the interglacial stages at ca. 1Mya of 256.3 ± 3.8 ppm (central upper blue bar Fig 1C), which is not significantly different to our predicted mean of 253.1 ± 2.3 ppm for the upper 25ᵗʰ percentile (hereafter interglacial) stages of the past 800 kyr (upper left blue bar Fig 1C). In comparison, the interglacial BI-CO₂ data from ca. 1Myr suggest a concentration of 271.3 ± 4.5 ppm, which although higher than the observed mean over the past 800 kyr of (253.9 ± 4.1 ppm, orange bar top left of Fig. 1C), is within the range of the observed interglacial CO₂ concentration during interglacials 5, 9 and 11 and is less than the glacial stage draw down suggested by the blue ice data."

> "The Higgins et al., (2015) BI- CO₂ data indicate greater glacial stage draw down than our predicted record. However, old ice from blue ice areas may be subject to diffusional smoothing of CO₂ [e.g. Yan *et al.,* 2019], which would act in the direction of elevating the minimum (lower 25ᵗʰ percentile and assumed glacial) values found in the blue ice above the glacial atmospheric values. Diffusion does not offer an explanation for the blue ice maxima inCO₂at 1Mya (upper 25ᵗʰ percentile and assumed interglacial) also being higher than our prediction, so here it may be that our model is under-predicting interglacial CO₂ values prior to the MPT. But under predication cannot be confirmed given the caveat that blue ice

samples, which are generally drawn from ice which has passed close to the base of the ice sheet, have risk of artificially elevated concentrations due in-situ respiration of detrital matter. Although respiration effects are screened for by measurements of δ13C of $CO_2$, it is difficult to demonstrate that all samples are unaffected (Yan et al., 2019). A further argument against rejecting our model predictions is comparison to the Yan et al., (2019) Allan Hills BI-$CO_2$ data from 1.5 ± 0.21 Mya; here we see our predicted interglacial and glacial $CO_2$ levels closely overlapping with the upper and lower 25th percentiles of the BI data (Figure R1-2).

The BOR-$CO_2$ data from Chalk et al., (2017) - like our prediction - also does not indicate any significant drawdown during interglacial stages. This is demonstrated by the upper 25 percentile mean for the early MPT BOR-$CO_2$ data (274.2 ± 13.4 ppm, green bar top right of Fig 1C), being not significantly different to the post MPT upper 25 percentile of 280.9 ± 14.8 ppm. Also like our predictions, the BOR-$CO_2$ data support significant glacial stage drawdown, as demonstrated by the lower 25 percentile mean for the early MPT data (238.7 ± 22.1 ppm, green bar bottom right Fig 1C), which is significantly higher than the post MPT lower 25 percentile mean (198.9 ± 10.4 ppm). We note that the post MPT interglacial BOR-$CO_2$ data exceed the observed concentration in the ice core record by ca. 26 ppm (Figure R1-3*) and that the early-MPT interglacial BOR-$CO_2$ exceed our predictions by a similar amount (Fig 1 C.). Therefore we conclude that the Chalk et al., BOR-$CO_2$ data also do not provide cause to reject our model predictions.

These changes would necessitate corresponding revisions to other parts of the manuscript, which we would undertake for resubmission.

[Figure]

*Figure R1-2: Predicted $CO_2$ (this work) and the Yan et al. (2019) blue ice $CO_2$ record from the Allan Hills. Black crosses represent the mean blue ice measurements at 1.5 ± 0.21 Mya filtered into the upper and lower 25th percentiles (with 2σ errors) to represent interglacial and glacial stages respectively and averaged over their age uncertainty range (210kyr).(*n.b. for a revised manuscript we would include this Yan et al. data in Fig 1.).*

[Figure]

*Figure R1-3: The Bereiter et al., (2015) observed 800 kyr CO₂ record (orange) and the Chalk et al. (2017) boron-isotope based CO₂ reconstruction (green) filtered into glacial and interglacial values via the upper and lower 25th percentile and averaged over the total time period of each record. This is compared to the predicted CO₂ from this study (blue curve) (\*n.b. for a revised manuscript we would consider including the Chalk et al., post MPT BOR-CO₂ data in revised Fig 2.).*

Line 185-186: The authors need to explain why out-of-phase responses in northern and southern ice before the MPT (as proposed by Raymo et al., 2006) would lead them to expect "large discrepancies" between their regression-based $CO_2$ prediction and the realized data. This inference seems to rely on the assumption of a certain relationship between $CO_2$ and northern or southern ice sheets, but I'm not sure what relationship the authors are assuming. Section 4.2 overall is quite short and would benefit from a more in-depth, process-based discussion of implications of the anti-phased hemisphere hypothesis for pre-MPT $CO_2$ variability.

Thank you, this is a good suggestion and we now also have the opportunity to further develop this section with reference to new work presented by Yan et al., 2022 on the phasing of northern and southern ice volume pre and post-MPT. We would further develop Section 4.2 with nominal text as follows:

"Previous work has shown that across the glacial–interglacial cycles captured in the Vostok and EPICA Dome C ice core records there is more than 80% common variance between observed atmospheric $CO_2$ and ice core water stable isotope-based reconstructions of Antarctic temperature [Cuffey and Vimeux, 2001; Wolff et al., 2005; Luthi et al., 2008)]. This observed correlation has contributed to a prevalent view that climate conditions in the circum-Antarctic Southern Ocean (which are assumed to be captured or at least correlate with the Antarctic temperature reconstruction) play a dominant role in modulating glacial-interglacial atmospheric $CO_2$ variations (see Fischer et al., 2010 for a review). The links between Southern Ocean conditions and atmospheric $CO_2$ remain contested, but are proposed to include climate-driven physical changes in $CO_2$ ventilation from the Southern Ocean associated with surface buoyancy (e.g. Watson and Garabato, 2006), sea ice variability as a cap to exchange (Stephens and Keeling, 2000), changes in wind-driven upwelling (e.g. Toggweiler et a., 2006) and temperature sensitivity of solubility (Millero 1995). In addition, there is much literature on direct and indirect modulation of biological carbon fluxes by the effects of SO climate conditions, including on SO export production of

organic material and carbonate compensation feedbacks in the deep ocean (Broecker and Peng, 1987; Fischer et al., 2010) and iron fertilisation (Martinez-Garcia et al., 2014). The assumption inherent in our predicted $CO_2$ record is that processes linking SO climate conditions, global ice volume and carbon cycle changes during the past 800 kyr can be extrapolated across the MPT. This assumption would be violated, and we would expect the model to fail, in the case that the phase locking hypothesis suggested by Raymo et al., (2006) holds. Some discussion of the basis of the phase locking hypothesis is required to understand why.

The phase locking hypothesis offers an explanation for the absence of precession-related (23kyr) periods in the LR04 benthic stack prior to the MPT (see Appendix Fig C), despite the strong precession cycle in insolation (Raymo et al., 2006, Morée et al., 2021). The key concept is that prior to the MPT the Northern Hemisphere and Antarctic ice sheets were responsive (in ice volume) to insolation changes in the precession band, but because precession forcing is out of phase between the hemispheres, the ice volume changes were opposing between the hemispheres and therefore cancelled in the benthic stack. This cancellation of the precession signal left insolation forcing in the 40 kyr obliquity band to dominate globally integrated ice volume changes expressed in the benthic stack. A transition from a smaller and more dynamic terrestrial terminating Antarctic ice sheet to a larger and more stable marine terminating ice sheet with cooling climate across the MPT (e.g. Elderfield et al., 2012) is then proposed to remove sensitivity of Antarctic ice volume to precession forcing and to suppress ice sheet sensitivity to the obliquity band in favour of quasi-100kyr ice volume changes that are in phase between the hemispheres (Raymo et al., 2006).

Recently presented data from Yan et al., support the phase locking hypothesis, specifically with evidence that pre-MPT Antarctic temperature (and by extension ice volume) is positively correlated with a local precession-band insolation proxy that is based on the oxygen-to-nitrogen ratio of trapped air (Yan et al., 2022). Whereas the correlation becomes negative in the blue ice and continuous ice core data post MPT.

To now extend the argument to potential impacts on $CO_2$ exchange, if the phase locking hypothesis holds, then prior to the MPT the Antarctic and Southern Ocean climate conditions and by extension the Southern Ocean mechanisms of $CO_2$ exchange described earlier, would also be expected to fall out of phase with the benthic stack. Since our regression model assumes continuation of the in-phase relationship between the benthic stack and Antarctic and Southern Ocean climate conditions (as inherited from the post-MPT training data) we would expect to see disagreement between our pre-MPT $CO_2$ predictions and a realised oldest ice continuous ice core $CO_2$ record."

**References (added in revisions)**

Archer, D., Winguth, A., D. Lea, and Mahowald, N.: What caused the glacial/interglacial atmospheric $pCO_2$ cycle?, *Rev. Geophys.*, **38**, 159–189, 2000, doi: 10.1029/1999RG000066, 2000.

Cuffey, K.M., Vimeux, F.: Covariation of carbon dioxide and temperature from the Vostok ice core after deuterium-excess correction. *Nature.*, **412**, 523–527, 2001

Fischer, H., Schmitt J., Lüthi, D., Stocker, T.F., Tschumi T., Parekh, P., Joos, F., Köhler, P., Völker, C.,, Gersonde, R., Barbante, C., Le Floch, M., Raynaud, D., and Wolff, E.: The role of Southern Ocean processes in orbital and millennial $CO_2$ variations – A synthesis, *Quaternary Science Reviews*, **29** (1–2), 193-205, 2010.

Gottschalk, J., Battaglia, G., Fischer, H., Frölicher, T.L., Jaccard, S.L., Jeltsch-Thömmes, A., Joos, F., Köhler, P., Meissner, K.J., Menviel, L., Nehrbass-Ahles, C., Schmitt, J., Schmittner, A., Skinner, L.C., and Stocker, T.G.: Mechanisms of millennial-scale atmospheric CO2 change in numerical model simulations, *Quaternary Science Reviews*, **220**, 30-74, https://doi.org/10.1016/j.quascirev.2019.05.013, 2019.

Martínez-García, A., Sigman, D.M., Ren, H., Anderson, R.F., Straub, M., Hodell, D.A., Jaccard, S.L., Eglinton, T.I., and Haug, G.H.: Iron fertilization of the subantarctic ocean during the last ice age, *Science*, **343** (6177), 1347-1350, doi: 10.1126/science.1246848, 2014.

Millero, F. J.: Thermodynamics of the carbon dioxide system in the oceans, *Geochimica et Cosmochimica Acta,* **59**, 661-677, 1995.

Morée, A. L., Sun, T., Bretones, A., Straume, E. O., Nisancioglu, K., and Gebbie, G.: Cancellation of the precessional cycle in $\delta^{18}O$ records during the Early Pleistocene. *Geophysical Research Letters*, **48**, e2020GL090035. doi: 10.1029/2020GL090035, 2021.

Shackleton, N. J. and Pisias, N. G.: Atmospheric Carbon Dioxide, Orbital Forcing, and Climate. In: The Carbon Cycle and Atmospheric CO2: Natural Variations Archean to Present, 1985.

Stephens, B.B., Keeling, R.F.: The influence of Antarctic sea ice on glacial–interglacial $CO_2$ variations. *Nature*, **404**, 171–174, 2000.

Toggweiler, J.R., Russell, J.L., and, Carson, S.R.: Midlatitude westerlies, atmospheric $CO_2$, and climate change during the ice ages. *Paleoceanography*, **21**, PA2005, doi: 10.1029/2005PA001154., 2006.

Ushie, H., and Matsumoto, K.: The role of shelf nutrients on glacial-interglacial $CO_2$: A negative feedback, *Global Biogeochem. Cycles*, 26, GB2039, doi:10.1029/2011GB004147., 2012.

Watson, A.J., Garabato, A.C.N.: The role of Southern Ocean mixing and upwelling in glacial–interglacial atmospheric $CO_2$ change. *Tellus*, **58B**, 73–87, 2006.

Yan, Y., Kurbatov, A.V., Mayewski, P.A. *et al.* Early Pleistocene East Antarctic temperature in phase with local insolation. *Nat. Geosci.*, doi: 10.1038/s41561-022-01095-x, 2022

---

## Author Comment (AC8)

An earlier incomplete version of the review response was uploaded on 24/11. Please refer to the final response below. Reviewer comments are in black text and our response in blue. We do not include a tracked changes version of the manuscript or a revised manuscript at this point as the editor request is to respond to the comments only and not to prepare a revised manuscript. We do provide in the response below edits that we would include in a revised manuscript.

*RC2*

An earlier incomplete version of the review response was uploaded on 24/11. Please refer to the final response below and the responses to the other reviews of our manuscript by R1 and Peter Kohler. Reviewer comments are in black text and our response in blue. We do not include a tracked changes version of the manuscript or a revised manuscript at this point as the editor request is to respond to the comments only and not to prepare a revised manuscript. We do provide in the response below edits that we would include in a revised manuscript.

*RC2* **Overall assessment**

The paper by Martin et al. represents a very simple, statistical (further on loosely called regression) model to estimate past atmospheric $CO_2$ from the LR05 stack of benthic $\delta^{18}O$. As LR05 is a combined record of deep ocean temperature and ocean volume (not of $CO_2$) the regression of $CO_2$ with LR05 is only statistical in nature and does not include a direct causal connection. Accordingly, a good predictive skill of LR05 to calculate $CO_2$ beyond its calibration period (the last 800 kyr) cannot be expected. Not surprisingly, the predicted $CO_2$ does not closely reflect the limited data we already have about $CO_2$ in the MPT from blue ice snap shots and $CO_2$ reconstructions based on $\delta^{11}B$ in foraminifera.

Based on this disagreement, the authors conclude that the null hypothesis of "a common global climate - carbon cycle - cryosphere feedback across the MPT" must be rejected. This is correct in a purely statistical sense, however, without laying out what exactly the causal relationship is between the three Earth System components and why these could be imprinted in the LR05/$CO_2$ regression, the null hypothesis appears to be not well justified. Accordingly, I think the minimum the author have to do to their manuscript is to discuss this connection and to bolster the justification of the null hypothesis. Another point of criticism could be raised that also the existing $CO_2$ from blue ice and $\delta^{11}B$ may contribute to the difference between observed and predicted $CO_2$. For example, the very old ice from the bottom of blue ice areas may be subject to diffusional smoothing of $CO_2$. This could explain that the minimum (glacial?) values found in the blue ice are higher than the true atmospheric values, however, it would not be in line with the (interglacial?) blue ice maxima in $CO_2$ being also higher than the prediction. Also the limits of the $\delta^{11}B$ reconstructions have to be better laid out as they are strongly dependent on the input parameters that are used to calculate $CO_2$ from $\delta^{11}B$ and also from the $CO_2$ saturation state at the marine drilling site in the past, as also illustrated by the relatively large uncertainty of the $\delta^{11}B$ reconstructions compared to ice core records.

In summary, while the study by Martin et al represents an interesting exercise (as was the initial EPICA challenge published in a non-peer reviewed journal), the question remains, whether this contribution in its present from provides sufficient new insight to justify publication in CP.

Thank you for the constructive critique. We believe the manuscript has been strengthened by addressing it. We make a number of changes to address the key concerns, summarised immediately below and more detail can be found further on in the response to specific comments.

Regarding the comment "..Not surprisingly, the predicted CO₂ does not closely reflect the limited data we already have about CO₂ in the MPT from blue ice snap shots and CO₂ reconstructions based on $\delta^{11}$B in foraminifera." In response, we bring in substantial additional data to evaluate our predicted record CO₂ record, specifically: Allan Hills Blue Ice CO₂ data at from Yan *et al.,* (2019) at 1.5 Mya; CO₂ proxy reconstructions from leaf wax Yamamoto *et al.,* (2022); $\delta^{11}$B reconstructions by Dyez *et al.,* (2018), and Guillermic *et al.,* (2022) and a high resolution CO₂ reconstruction by van de Wal *et al.* (2011). The comparison is shown in a revised Fig 2, below. In the revised figure we see that our predicted CO₂ sits around the middle of the field compared to other estimates and is consistent with the Yan et al (2019) data. Accordingly, we now consider that the null hypothesis cannot yet be rejected by the existing data. Detailed quantitative comparisons between our predicted CO₂ and the existing data are provided in response to comments further below and in the response to R1.

[Figure]

Figure 2 (revised): Predicted CO₂ (this work) and observed, proxy and modelled CO₂ from a range of other sources: $\delta^{11}$B-based pCO₂ reconstructions and measurements by Dyez *et al.* (2018), Guillermic *et al.* (2022) & Chalk *et al.* (2017); model simulation under a regolith removal hypothesis by Willeit *et al.* (2019); blue ice CO₂ measurements by Yan *et al.* (2019) & Higgins *et al.* (2015); $\delta$13C leaf wax proxy reconstructions by Yamamoto *et al.* (2022); high resolution CO₂ reconstruction by van de Wal *et al.* (2011)

Regarding the comment "Accordingly, I think the minimum the author have to do to their manuscript is to discuss this (benthic stack to CO₂) connection and to bolster the justification of the null hypothesis." In response, we propose to add sections in the introduction and discussion discussing potential direct and indirect physical and/or biogeochemical links between LR04 global ice volume and temperature proxy and atmospheric CO₂. The proposed new text is provided further below in the response to specific comments.

We also propose to add discussion of the caveats associated with blue ice and marine sediment boron-isotope based CO₂ reconstructions, again given in full further below.

Fully updating the manuscript to capture all the flow on changes is beyond the scope of what is requested by the editor at this stage, but we hope the responses below now better demonstrate the potential value and suitability of the work for a revised submission to CP.

**R2 Specific comments:**

***R2 Comment line 16:*** *"is to make"*

Accepted.

***R2 Comment:*** *line 17 and throughout the manuscript: Myr instead of myr*

Accepted.

***R2 Comment:*** *line 25: the authors state that the null hypothesis should be rejected, however, without laying out the causal relationship between the regression parameters and potential reasons why the regression may not hold back in time, this statement is not entirely satisfying.*

We make substantial revisions in three main areas to address this comment.

First, we add more detail on potential mechanisms linking the regression parameters, with nominal new text added around Lines 58-59 as below:

This is a good suggestion and in response we would revise the manuscript (nominally around lines 58-59) to give more detail on potential underlying mechanisms (while also acknowledging that there is no consensus on these):

> "The $\delta^{18}O$ of fossil benthic foraminifera calcite is governed by ocean temperature and global ice volume at the time the foraminifera lived, with higher values indicating both increased ice sheet volume and a colder climate. The first order rationale in using the LR04 $\delta^{18}O$ benthic stack as an input parameter to predict $CO_2$ is based on the known relationship of ocean temperature with $CO_2$ solubility (e.g. Millero, 1995). The solubility of $CO_2$ in the ocean increases with falling sea surface temperatures, particularly in high-latitude deep-water formation regions, where colder ocean temperature drive increased uptake of $CO_2$ by the ocean, reducing the atmospheric $CO_2$ concentration (Berends et al., 2021). However, we note that the magnitude of glacial cooling can only account for a portion of observed glacial $CO_2$ drawdown (Archer et al., 2000) and multiple other dependencies between $CO_2$ and ocean temperature and/or ice volume are also likely at play in explaining the observed shared variance. These may include (not exhaustively), direct radiative forcing of ice volume changes by $CO_2$ (e.g. Shackleton et al., 1985), second order effects on atmospheric $CO_2$ from changing ice volume, including the impact of ice volume/sea level on atmospheric $CO_2$ via ocean productivity and carbonate chemistry changes (e.g. Broecker et al., 1982; Archer et al., 2004; Ushie and Matsumoto 2012), $CO_2$ drawdown during periods of high ice volume by increased iron fertilization (e.g. Rothlisberger et al., 2004; Martinez-Garcia et al., 2014) and enhanced sea ice extent during periods of high ice volume capping the ventilation of $CO_2$ from the ocean interior at high latitudes (Stephens and Keeling, 2000)."

And we would add corresponding text to the discussion around Lines 177, nominally:

> "The range of physical and biogeochemical processes involved in glacial to interglacial $CO_2$ changes are still contested and glacial to interglacial $CO_2$ source and sink changes are still not quantitatively constrained (Sigman et al., 2010; Fischer et al., 2010). Given the fundamental uncertainties we cannot quantitatively set out and apportion physical processes responsible for the observed regression. Nevertheless, our analysis shows that 68% of the variance in observed $CO_2$ over the past 800 kyr is common with the LR04 stack and that strength of this relationship is remarkably insensitive to our bootstrap tests that remove 50% of the data (Methods)."

In addition we develop the discussion around potential reasons why our regression model may not hold prior to the MPT. As an example, we would expand the discussion on the phase locking hypothesis (which was recently bolstered by the new data from Yan et al., 2022). Nominal additions to Section 4.2 as follows:

"Previous work has shown that across the glacial–interglacial cycles captured in the Vostok and EPICA Dome C ice core records there is more than 80% common variance between observed atmospheric $CO_2$ and ice core water stable isotope-based reconstructions of Antarctic temperature [Cuffey and Vimeux, 2001; Wolff et al., 2005; Luthi et al., 2008)]. This observed correlation has contributed to a prevalent view that climate conditions in the circum-Antarctic Southern Ocean (which are assumed to be captured or at least correlate with the Antarctic temperature reconstruction) play a dominant role in modulating glacial-interglacial atmospheric $CO_2$ variations (see Fischer et al., 2010 for a review). The links between Southern Ocean conditions and atmospheric $CO_2$ remain contested, but are proposed to include climate-driven physical changes in $CO_2$ ventilation from the Southern Ocean associated with surface buoyancy (e.g. Watson and Garabato, 2006), sea ice variability as a cap to exchange (Stephens and Keeling, 2000), changes in wind-driven upwelling (e.g. Toggweiler et a., 2006) and temperature sensitivity of solubility (Millero 1995). In addition, there is much literature on direct and indirect modulation of biological carbon fluxes by the effects of SO climate conditions, including on SO export production of organic material and carbonate compensation feedbacks in the deep ocean (Broecker and Peng, 1987; Fischer et al., 2010) and iron fertilisation (Martinez-Garcia et al., 2014). The assumption inherent in our predicted $CO_2$ record is that processes linking SO climate conditions, global ice volume and carbon cycle changes during the past 800 kyr can be extrapolated across the MPT. This assumption would be violated, and we would expect the model to fail, in the case that the phase locking hypothesis suggested by Raymo et al., (2006) holds. Some discussion of the basis of the phase locking hypothesis is required to understand why.

The phase locking hypothesis offers an explanation for the absence of precession-related (23kyr) periods in the LR04 benthic stack prior to the MPT (see Appendix Fig C), despite the strong precession cycle in insolation (Raymo et al., 2006, Morée et al., 2021). The key concept is that prior to the MPT the northern hemisphere and Antarctic ice sheets were responsive (in ice volume) to insolation changes in the precession band, but because precession forcing is out of phase between the hemispheres, the ice volume changes were opposing between the hemispheres and therefore cancelled in the benthic stack. This cancellation of the precession signal left insolation forcing in the 40 kyr obliquity band to dominate globally integrated ice volume changes expressed in the benthic stack. A transition from a smaller and more dynamic terrestrial terminating Antarctic ice sheet to a larger and more stable marine terminating ice sheet with cooling climate across the MPT (e.g. Elderfield et al., 2012) is then proposed to remove sensitivity of Antarctic ice volume to precession forcing and supress too ice sheet sensitivity to the obliquity band in favour of quasi-100kyr ice volume changes that are in phase between the hemispheres (Raymo et al., 2006).

Recently presented data from Yan et al., support the phase locking hypothesis, specifically with evidence that pre-MPT Antarctic temperature (any by extension ice volume) is positively correlated with a local precession-band insolation proxy that is based on the oxygen-to-nitrogen ratio of trapped air (Yan et al., 2022). Whereas the correlation becomes negative in the blue ice and continuous ice core data post MPT.

To now extend the argument to potential impacts on $CO_2$ exchange, if the phase locking hypothesis holds, then prior to the MPT the Antarctic and Southern Ocean climate conditions and by extension the Southern Ocean mechanisms of $CO_2$ exchange described earlier, would also be expected to fall out of phase with the benthic stack. Since our regression model assumes continuation of the in-phase relationship between the benthic stack and Antarctic and Southern Ocean climate conditions (as inherited from the post-MPT training data) we would expect to see disagreement between our pre-MPT $CO_2$ predictions and a realised oldest ice continuous ice core $CO_2$ record."

**R2 Comment:** *line 58-59: $\delta^{18}O$ is not just a sea level proxy but also influenced by deep ocean temperature. A process-based discussion of why LR05 is a viable input parameter to predict $CO_2$ is required.*

Addressed within the response to the earlier R2 comment on the physical basis of the regression.

**R2 Comment:** *Line 66: please include also the record by Dyez et al., Paleoceanography 2018:*

Thanks for the suggestion. This comment refers to the Dyez et al. (2018) $\delta^{11}B$-based $pCO_2$ proxy data spanning 1.38–1.54 Ma. We will add it to the description of records around Line 66, as follows:

"To test the null hypothesis, we compare our predicted $CO_2$ record to several existing sets of $CO_2$ or $CO_2$ proxy data that exist within the predicted range: 1) $CO_2$ estimates from the analysis of boron isotope ratios in benthic sediment cores which present a proxy for ocean pH to which a transfer function is applied to reconstruct atmospheric $CO_2$ (hereafter referred to as BOR-$CO_2$) (Chalk, et al., 2017; Henehan et al., 2013; Dyez et al., 2018).."

This record plotted is plotted below in Figure R2-1 for comparison with our predicted $CO_2$ record.

[Figure]

*Figure R2-1: Predicted $CO_2$ (this work) and the Dyez et al. (2018) $\delta^{11}B$-based $pCO_2$ reconstruction.*

We would include new text in the discussion along lines:

"For the period of overlap with our predicted record, 1.38–1.5 Mya, the Dyez et al. (2018) $\delta^{11}B$-based $pCO_2$ data appears higher during interglacial periods by 20 to 50 ppm. However the significance of this difference is questionable given the ca. ±33 ppm uncertainty in the

δ[11]B-based data. What is clearer, is that there is no significant difference between our predicted record and the Dyez et al., data during glacial stages."

We will also revise Figure 2 in the main manuscript to include the Dyez et al., data (see the revised Fig 2 in our response to the general comments). Including comparisons to this and a number of other records suggested in this and other reviews leads us to revisit our earlier conclusion that our predicted $CO_2$ record increasingly under predicts $CO_2$ concentrations indicated by other archives. Rather, our predicted record sits around the central range of other predictions over the 0.8 to 1.5Mya interval.

***R2 Comment,*** *line 68: The very old ice at Allan Hills is not really from the surface but from a shallow ice drilling of more than 100 m depth.*

Good point. Text adjusted as follows:

"..glacial ice that has been brought to the near surface of an ice sheet by ice flow processes, where it is can be accessed by cutting trenches or by shallow drilling of up to several hundred meters (e.g. Higgins et al., 2015). "

***R2 Comment:*** *Methods: the uncertainty in the regression connected to the independent age scales should be discussed*

Good point. We would add some discussion in the revised Methods:

"Limitations in the regression may exist due to uncertainties in the independent age scales of both the LR04 stack and the compiled $CO_2$ record, even after binning to the 3 kyr grid. The LR04 stack graphically correlates 57 globally distributed δ[18]O sediment cores through common climate signals with an independently developed age model constructed from the average sedimentation rates of each core, assuming global sedimentation rates have remained relatively stable, and tuned to a simple ice model. The authors estimate uncertainty of 6 kyr from 1.5 – 1.0 Mya, and 4 kyr from 1 – 0 Mya due to the tuning technique neglecting higher frequency changes over global climate reorganisations and glacial cycles (Lisiecki & Raymo, 2005). The composite observed $CO_2$ ice core record (Bereiter at al., 2015) uses six independent dating methods for various core locations both spatially across Antarctica, and stratigraphically for different sections of the same core (firn, ice etc.). The age uncertainty in the gas timescale has a median over the 0—800kyr interval of 2 kyr, but individual uncertainties can reach up to 5 kyr (Veres et al 2013; Bazin et al., 2013). The relative age uncertainties between these input variables may diminish the regression or in some instances lead to spurious correlation. However, we expect any such effects are minor on the basis that our predictions show little sensitivity (Fig 1A) to the bootstrap analysis with 1000 iterations of recomputing the regression after removing 50% of data (Methods). If the model was sensitive to relative timescale uncertainties we would expect larger bootstrap confidence intervals."

***R2 Comment:*** *line 85: not clear what r(226) means, please explain. Did you allow for lag correlation? (see also comment on age scales above)*

We were referring here to the correlation coefficient between our predicted record of $CO_2$ and the observed composite ice core record. '226' refers to the degrees of freedom of the test. We change

this to simply report the Pearson correlation coefficient, in this case r = -0.82 (p = <<0.01). Our generalised least square model accounted for autocorrelation/lag between the predictor ($\delta^{18}$O) and $CO_2$ using an AR(1) correlation factor.

**R2 Comment:** line 89: the limitations of blue ice $CO_2$ reconstructions and $\delta^{11}$B reconstructions of $CO_2$ should be discussed as well.

We plan to add further detail including around line 130 on blue ice and $\delta^{11}$B limitations in the opening of the discussion:

> "$CO_2$ measurements from blue ice, and proxy-reconstructions from boron isotopes have a number of caveats. Old ice from blue ice areas may be subject to diffusional smoothing of $CO_2$ (Yan et al., 2019) and deformation (Higgins et al., 2015), which would act in the direction of elevating the minimum (lower 25th percentile and assumed glacial) values found in the blue ice higher above the glacial atmospheric values. A caveat that is difficult to entirely eliminate with blue ice samples (which are generally drawn from ice which has passed close to the base of the ice sheet) is the potential for elevated $CO_2$ levels due to contamination by in-situ respiration of detrital matter. Although respiration effects are screened for by measurements of δ13C of $CO_2$, it is difficult to demonstrate that all samples are unaffected (Yan et al., 2019)…
>
> … Further, $\delta^{11}$B is strongly dependent on the input parameters that are used to calculate $CO_2$ from $\delta^{11}$B, i.e., reconstruction of past, local marine carbon chemistry for which additional components of the carbonate system must be known or assumed (Hönisch et al., 2009). $\delta^{11}$B is also dependent on the $CO_2$ saturation state at the marine drilling site in the past, contributing to the relatively large uncertainty (10s of ppm) of the $\delta^{11}$B reconstructions compared to continuous ice core records (several ppm)). However, despite the large uncertainties, atmospheric $CO_2$ reconstructions from boron isotopes by Chalk et al. (2017) from ~0 - 250 kyr displays consistency with the observed ice core record in glacial stages over the last 800 kyr when filtered by the lower 25th percentile of the respective $\delta^{18}$O values and averaged over their common time intervals (198.9 ± 10.4 and 202.0 ± 3.2 ppm respectively). However, interglacial stage $CO_2$ over this time is not consistent between the boron based estimates and the observed 800 kyr record (280.9 ± 14.8 and 253.9 ± 4.1 ppm. respectively) indicating that the boron based estimates are recording artificially elevated $CO_2$, at least for the interglacial stages.

[Figure]

*Figure R2-3: Predicted* $CO_2$ *(this work, blue curve) compared with the Bereiter et al., (2015) observed 800 kyr* $CO_2$ *record filtered into glacial and interglacial values via the upper and lower 25th percentile and averaged over the total time and the boron-isotope based* $CO_2$ *reconstruction from Chalk et al., (2017), filtered into glacial and interglacial values via the upper and lower 25th percentile and averaged over the total time period. (\*n.b. for a revised manuscript we would consider including the Chalk et al., post MPT BOR-$CO_2$ data in revised Fig 2.).*

**References (added in revisions)**

Archer, D., Winguth, A., D. Lea, and Mahowald, N.: What caused the glacial/interglacial atmospheric $pCO_2$ cycle?, *Rev. Geophys.*, **38**, 159–189, 2000, doi: 10.1029/1999RG000066, 2000.

Cuffey, K.M., Vimeux, F.: Covariation of carbon dioxide and temperature from the Vostok ice core after deuterium-excess correction. *Nature.*, **412**, 523–527, 2001

Dyez, K.A., Hönisch, B., and Schmidt, G.A.: Early Pleistocene obliquity-scale $pCO_2$ variability at ~1.5 million years ago. *Paleoceanogr. Paleoclimatol.*, **33**, no. 11, 1270-1291, doi:10.1029/2018PA003349, 2018.

Fischer, H., Schmitt J., Lüthi, D., Stocker, T.F., Tschumi T., Parekh, P., Joos, F., Köhler, P., Völker, C.,, Gersonde, R., Barbante, C., Le Floch, M., Raynaud, D., and Wolff, E.: The role of Southern Ocean processes in orbital and millennial $CO_2$ variations – A synthesis, *Quaternary Science Reviews*, **29** (1–2), 193-205, 2010.

Martínez-García, A., Sigman, D.M., Ren, H., Anderson, R.F., Straub, M., Hodell, D.A., Jaccard, S.L., Eglinton, T.I., and Haug, G.H.: Iron fertilization of the subantarctic ocean during the last ice age, *Science*, **343** (6177), 1347-1350, doi: 10.1126/science.1246848, 2014.

Millero, F. J.: Thermodynamics of the carbon dioxide system in the oceans, *Geochimica et Cosmochimica Acta,* **59**, 661-677, 1995.

Morée, A. L., Sun, T., Bretones, A., Straume, E. O., Nisancioglu, K., and Gebbie, G.: Cancellation of the precessional cycle in $\delta^{18}O$ records during the Early Pleistocene. *Geophysical Research Letters*, **48**, e2020GL090035. doi: 10.1029/2020GL090035, 2021.

Shackleton, N. J. and Pisias, N. G.: Atmospheric Carbon Dioxide, Orbital Forcing, and Climate. In: The Carbon Cycle and Atmospheric CO2: Natural Variations Archean to Present, 1985.

Stephens, B.B., Keeling, R.F.: The influence of Antarctic sea ice on glacial–interglacial $CO_2$ variations. *Nature*, **404**, 171–174, 2000.

Toggweiler, J.R., Russell, J.L., and, Carson, S.R.: Midlatitude westerlies, atmospheric $CO_2$, and climate change during the ice ages. *Paleoceanography*, **21**, PA2005, doi: 10.1029/2005PA001154., 2006.

Ushie, H., and Matsumoto, K.: The role of shelf nutrients on glacial-interglacial $CO_2$: A negative feedback, *Global Biogeochem. Cycles*, 26, GB2039, doi:10.1029/2011GB004147., 2012.

Bazin, L., Landais, A., Lemieux-Dudon, B., Toye Mahamadou Kele, H., Veres, D., Parrenin, F., Martinerie, P., Ritz, C., Capron, E., Lipenkov, V., Loutre, M.-F., Raynaud, D., Vinther, B., Svensson, A., Rasmussen, S., Severi, M., Blunier, T., Leuenberger, M., Fischer, H., Masson-Delmotte, V., Chappellaz, J., and Wolff, E.: An optimized multi-proxies, multi-site Antarctic ice and gas orbital chronology (AICC2012): 120-800 ka, *Climate of the Past*, **9**, 1715-1731, doi:10.5194/cp-9-1715-2013, 2013.

Veres, D., Bazin, L., Landais, A., Toye Mahamadou Kele,H., Lemieux-Dudon, B., Parrenin, F., Martinerie, P., Blayo, E., Blunier, T., Capron, E., Chappellaz, J., Rasmussen, S., Severi, M., Svensson, A., Vinther, B., and Wolff, E.: The Antarctic ice core chronology (AICC2012): an optimized multi-parameter and multi-site dating approach for the last 120 thousand years, *Climate of the Past*, **9**, 1733-1748, doi:10.5194/cp-9-1733-2013, 2013.

Watson, A.J., Garabato, A.C.N.: The role of Southern Ocean mixing and upwelling in glacial–interglacial atmospheric $CO_2$ change. *Tellus*, **58B**, 73–87, 2006.

Yan, Y., Kurbatov, A.V., Mayewski, P.A. *et al.* Early Pleistocene East Antarctic temperature in phase with local insolation. *Nat. Geosci.*, doi: 10.1038/s41561-022-01095-x, 2022

---

## Author Comment (AC9)

An earlier incomplete version of the review response was uploaded on 24/11. Please refer to the final response below and the responses to the other reviews of our manuscript. Reviewer comments are in black text and our response in blue. We do not include a tracked changes version of the manuscript or a revised manuscript at this point as the editor request is to respond to the comments only and not to prepare a revised manuscript. We do provide in the response below edits that we would intend to include in a revised manuscript.

***Peter Kohler***

This is a potentially interesting study, which might gain from some more discussions of what has already been done with respect to $CO_2$ across the MPT. Some comments, which might be of interest to the authors:

This was a very helpful review, many thanks Peter for taking the time. The main change in response is to add new comparisons and discussion of the additional $CO_2$ observations and proxy data suggested and to further develop our discussion of the null hypothesis, which as a result we now consider cannot be rejected. We will provide a revised Figure 2 with the suggested records included, similar to below. More detailed discussion of the comparisons in the responses below and also in the responses to R1 and R2.

[Figure]

*Figure 2 (revised): Predicted $CO_2$ (this work) and observed, proxy and modelled $CO_2$ from a range of other sources: $\delta^{11}B$-based $pCO_2$ reconstructions and measurements by Dyez et al. (2018), Guillermic et al. (2022) & Chalk et al. (2017); model simulation under a regolith removal hypothesis by Willeit et al. (2019); blue ice $CO_2$ measurements by Yan et al. (2019) & Higgins et al. (2015); δ13C leaf wax proxy reconstructions by Yamamoto et al. (2022); high resolution $CO_2$ reconstruction by van de Wal et al. (2011).*

**1.** To be transparent in what has been done, the equation which calculates $CO_2$ out of the LR04 benthic $\delta^{18}O$ stack is missing. Plotting of the LR04 benthic $\delta^{18}O$, which is at the core of the approach is also missing.

We will include the form of the equation as:

$$CO_2 = -33.37 \times \delta^{18}O + 365.16, \text{ autoregressive correlation factor (AR): } 1$$

We will also include the LR04 stack in Fig 1.

**2.** Blue ice CO$_2$ data from Allan Hills have been extended in Yan et al (2019), now also containing snapshots of CO$_2$ at 1.5 and 2.0 Ma.

Excellent. We will include the Yan et al., 2019 data in a revised Figure 1 and in Figure 2. The data shown against our prediction is shown below (Fig R3-2). We would include new text on the resulting comparison:

> "A further argument against rejecting our model predictions is comparison to the Yan et al., (2019) Allan Hills BI-CO$_2$ data from 1.5 ± 0.21 Mya; here we see our predicted interglacial and glacial CO$_2$ levels closely overlapping with the upper and lower 25$^{th}$ percentiles of the BI data (Figure R1-2)."

[Figure]

*Figure R3-2: Predicted CO$_2$ (this work) and the Yan et al. (2019) blue ice CO$_2$ record from the Allan Hills. Black crosses represent the mean blue ice measurements at 1 Mya filtered into the upper and lower 25th percentiles (with 2σ errors) to represent interglacial and glacial stages respectively and averaged over their age uncertainty range (210kyr).*

3. A recent paper by Yamamoto et al (2022) calculates CO$_2$ over the MPT from leaf wax d13C and finds that smaller glacial/interglacial amplitudes in CO$_2$ before the MPT are based on stable glacial CO$_2$, but smaller interglacial CO$_2$ before the MPT. This differs to the δ$^{11}$B-based CO$_2$, and if I got it right might support the here defined Null Hypothesis, which then cannot easily be dismissed.

Agree. The Yamamota data is shown against our predictions below (Fig R3-3). Their pre-MPT reconstruction trends below ours (and other observations) for glacial and interglacial stage CO$_2$. On the basis of this and the Yan et al (2019) BI-CO$_2$ data we can no-longer confidently reject the null hypothesis and will adjust the manuscript accordingly. In our view it makes for the more interesting that our simple GLS model cannot yet be rejected by the available data.

[Figure]

*Figure R3-3: Predicted CO₂ (this work) and the Yamamoto et al. (2022) leaf wax-based proxy CO₂ record.*

**4.** New CO₂ data based on δ11OB from Pacific cores have recently been published (Guillermic et al., 2022). Ok, data coverage across the last 1.5Ma might be weak, but worth discussing it.

You're right, average coverage across the MPT is not enough to filter into G/IG averages as we have done with δ11ON and the blue ice and many CO₂ values appear implausibly large. But we include the data in the revised Figure 2 and close up view below (Fig R3-4).

[Figure]

*Figure R3-4: Predicted CO₂ (this work) and the Guillermic et al., 2022 δ11OB-based CO₂ data from Pacific marine sediment cores.*

**5.** CO₂ as function of benthic δ¹⁸O has in an inverse modelling approach already been calculated by Stap et al (2016). This approach has been updateded by Berends et al. (2021a). So comparison to their results might tell, how (if at all) this study shows something new.

Our CO₂ prediction, like Berends et., 2021a was both trained on data from the recent 800 kyr and motivated by comparison to the upcoming oldest ice core records. Our simple model yields a high correlation to the observed 800 kyr Bereiter et al., CO₂ record  (r2 0.68) and our CO₂ predictions out

to 1.5 Myr can not be confidently excluded by the available blue ice and $CO_2$ proxy reconstructions. From what I understand Berends et., 2021a does not make any evaluation or comparison to the discrete δ11OB and blue ice data over the MPT.

**6.** Maybe also discuss other approaches of $CO_2$ across the MPT, eg C cycle simulation results (apart from those in Willeit et al, 2020, which are cited) of Köhler & Bintanja (2006), or the compilation of at that time available $CO_2$ data and the calculation of a continous high-resolution $CO_2$ record in van de Wal et al. (2011), updated in Stap et al. (2018).

We are happy to include discussions of Kolhler and Bintanja (2006); from our understanding the paper also creates a model based on the LR04 benthic stack as a null hypothesis, which sets precedence to our method. We originally used the model by Willeit et al., as an example of a model-based trajectory in which $CO_2$ departs from the LR04 based predictions. We will include the data from van de Wal in Figure 2 and see close up comparison with our predicted $CO_2$ in Fig R-5 below.

[Figure]

*Figure R3-5: Predicted $CO_2$ (this work) and an alternative prediction from van de Wal et al., 2011.*

7. The recent review on the MPT (Berends et al., 2021b) gives also an idea about processes including a collection of $CO_2$ data and discusses a potential influence of the carbon cycle on the climate transition.

Thank you. In response also to R1 and R2 we include a lot of new material and references to prior work on the potential physical basis of the regression between the LR04 stack and atmospheric $CO_2$ including treatment of the phase locking (or sometimes 'Antiphase') hypothesis of Raymo et al., 2006 which could alter the nature of the Southern Ocean contribution to $CO_2$ variability with respect to the timing of ice volume changes in the northern hemisphere ice sheets. Please refer to the response to these reviews. We will add reference to Berends et al., and include additional discussion of other proposed carbon cycle influences on the climate/ice volume across the MPT.

8. While mentioning the call for the EPICA challenge, maybe also cite / discuss its results (Wolff et al., 2005). They have been shown on 2 posters at AGU fall meeting in 2004 (PDFs for download at: https://epic.awi.de/id/eprint/11721/, https://epic.awi.de/id/eprint/11722/), on which you see, that

one of the participants to the challenge (N Shackleton) also used $\delta^{18}O$ to predict $CO_2$ for the 400-800 ky time window.

Thank you. We will add references and discuss the precedence in using $\delta^{18}O$ to predict $CO_2$ by N. Shackleton and add references to Berends and van de Wal around lines 61:

> "The use of benthic $\delta^{18}O$ to predict atmospheric $CO_2$ has precedence. In response to the EPICA challenge (Wolff et al., 2004), N. Shackleton (EGU, 2004) used this method to predict atmospheric $CO_2$ out to 800 kyr. Furthermore, inverse modelling of $CO_2$ using forced by the LR04 benthic stack has been undertaken by Berends et al. (2021a) and van de Wal et al. (2011)."

References:

van de Wal, R. S. W., de Boer, B., Lourens, L. J., Köhler, P., and Bintanja, R.: Reconstruction of a continuous high-resolution CO2 record over the past 20 million years. *Climate of the Past*, **7**, 1459–1469. doi:10.5194/cp-7-1459-2011, 2011.

Berends, C. J., de Boer, B., and van de Wal, R. S. W.: Reconstructing the evolution of ice sheets, sea level, and atmospheric CO2 during the past 3.6 million years. *Climate of the Past*, **17**, 361–377. doi:10.5194/cp-17-361-2021, 2021a

Berends, C. J., Köhler, P., Lourens, L. J., and van de Wal, R. S. W.: On the cause of the mid-Pleistocene transition., *Reviews of Geophysics*, **59**, e2020RG000727. doi:10.1029/2020RG000727, 2021b.

Guillermic, M., Misra, S., Eagle, R., and Tripati, A.: Atmospheric CO2 estimates for the Miocene to Pleistocene based on foraminiferal $\delta^{11}B$ at Ocean Drilling Program Sites 806 and 807 in the Western Equatorial Pacific, *Climate of the Past*, **18(2)**, 183–207, doi:10.5194/cp-18-183-2022, 2022.

Köhler, P., and Bintanja, R.: The carbon cycle during the Mid Pleistocene Transition: the Southern Ocean Decoupling Hypothesis, *Climate of the Past*, **4**, 311–332, doi:10.5194/cp-4-311-2008, 2008

Stap, L. B., de Boer, B., Ziegler, M., Bintanja, R., Lourens, L. J., and van de Wal, R. S. W.: CO2 over the past 5 million years: Continuous simulation and new $\delta^{11}B$-based proxy data., *Earth and Planetary Science Letters*, **439**, 1 – 10, doi: 10.1016/j.epsl.2016.01.022, 2016.

Stap, L. B., van de Wal, R. S. W., de Boer, B., Köhler, P., Hoencamp, J. H., and Lohmann, G., et al.: Modeled influence of land ice and CO2 on polar amplification and paleoclimate sensitivity during the past 5 million years. *Paleoceanography and Paleoclimatology*, **33**, 381–394. doi:10.1002/2017pa003313, 2018.

van de Wal, R. S. W., de Boer, B., Lourens, L. J., Köhler, P., and Bintanja, R.: Reconstruction of a continuous high-resolution CO2 record over the past 20 million years. *Climate of the Past*, **7**, 1459–1469. https://doi.org/10.5194/cp-7-1459-2011, 2011.

Wolff, E. W.; Kull, C.; Chappellaz, J.; Fischer, H.; Miller, H.; Stocker, T. F.; Watson, A. J.; Flower, B.; Joos, F.; Köhler, P.; Matsumoto, K.; Monnin, E.; Mudelsee, M.; Paillard, D., and, Shackleton, N. Modeling past atmospheric CO2: results of a challenge EOS, 86 (38), 341, 345, doi: 10.1029/2005EO380003, 2005.

Yamamoto, M., Clemens, S.C., Seki, O., Tsuchiya, Y., Huang, Y., O'ishi, R., and Abe-Ouchi, A.: Increased interglacial atmospheric CO2 levels followed the mid-Pleistocene Transition, *Nature Geoscience*, **15(4)**, 307–313, doi: 10.1038/s41561-022-00918-1, 2022.

Yan, Y., Benderm M.l., Brook, E.J., Clifford, H.M., Kemeny, P.C., Kurbatov, A.V., Mackay, S., Mayewski, P.A., Ng, J., Severinghaus J.P., and Higgins, J.A.: Two-million-year-old snapshots of atmospheric gases from Antarctic ice, *Nature*, **574(7780)**, 663–666, doi:10.1038/s41586-019-1692-3, 2019.

---

## Author Response (AR1)

*Predicting trend in atmospheric carbon dioxide using existing climate archives.*
POST REVISION REVIEW RESPONSES
JRW. Martin, J. Pedro, T. Vance

We thank the reviewers for their valuable suggestions on our paper. In response, we have significantly revised this manuscript. As requested, we have greatly increased the number of external data sources used to compare to our simple GLS-based carbon dioxide reconstruction over the mid Pleistocene transition and addressed all other reviewer suggestions. Following these revisions, we find no clear evidence to reject our predicted carbon dioxide record and inherent null hypothesis. This combined review response should be read together with our earlier responses to the individual reviews, as requested by the Copernicus editorial team. The revised manuscript was prepared after completing the earlier individual review responses. This combined review response contains the specific changes to the manuscript in response to the reviews, whilst the earlier review responses were more general.

**RC1:**

**Comment:** *"The analysis appears to be performed well, and I have only a few concerns about the interpretation of the results. However, my main concern is that the work is too simple. I would encourage the authors to add more intellectual substance to the paper by exploring perhaps nonlinear regression between benthic $\delta^{18}O$ and $CO_2$".*

Our objective with this manuscript was to generate the simplest reasonable model to predict $CO_2$ from the $\delta^{18}O$ benthic stack and to test the predictions against available observations. In response to this review and comments from other reviews we have substantially expanded the work in three main areas:

1. We bring in additional data to evaluate our predicted record $CO_2$, specifically: Allan Hills Blue Ice $CO_2$ data from Yan et al., (2019) at 1.5 Mya; $CO_2$ proxy reconstructions from leaf wax from Yamamoto et al., (2022); $\delta^{11}B$ reconstructions by Dyez et al., (2018), and Guillermic et al., (2022) and a high-resolution $CO_2$ reconstruction by van de Wal et al. (2011).

   See revised fig. 03 (Line 231), and revised fig. 04 (Line 337), which show these records. Further, additional text has been added:

   - The Results section (line 193-262) now primarily focuses on the two blue ice data sets; specifically, their comparison to each other, the current continuous $CO_2$ records, and to our modelled predictions.
   - Additional text on other proxy-based, and model-based estimates of $CO_2$ across the MPT is now provided in the Discussion (see manuscript lines 299-335).

2. In light of the additional comparisons with blue and proxy $CO_2$ records from (1.) we find that our regression-based prediction of pre-MPT $CO_2$ is not rejected by the existing data and we discuss the implications (see Discussion lines 322 to 389). Further, see line 266-285 for additional justification and discussion of our use of the GLS regression model over more complex, non-linear alternatives:

"It is possible that the fit between observed and our predicted $CO_2$ data could be further improved using a non-linear approach. However, we refrain from a non-linear approach for several key reasons. First, a scatter plot of the LR04 $\delta^{18}O$ benthic stack versus observed ice core $CO_2$ over the past 800 kyr yields a Pearson's correlation coefficient (R) of -0.82 (Fig. 2), indicating that ~68% of the variance in observed $CO_2$ is shared with the benthic stack. Importantly, there is no evidence in this scatter plot for departure from the linear relationship at high or low $CO_2$ or benthic $\delta^{18}O$ levels. Second, following the approach of Chalk *et al*., 2017 and interpreting the upper 25[th] percentile of $CO_2$ data as representing mean interglacial stage $CO_2$ and the lower 25[th] percentile of $CO_2$ data as representing mean glacial stages $CO_2$ levels, we see that our predicted interglacial mean value for the past 800 kyr (253.1 ± 2.3 ppm) closely overlaps with the observed interglacial mean value (253.9 ± 4.1 ppm) and similarly, the predicted glacial stage mean (199.7 ± 1.7 ppm) closely overlaps with the observed glacial stage mean (202.0 ± 3.2 ppm). Third, the predictions are remarkably insensitive to bootstrap analysis in which 50 % of that data are omitted with each iteration of the GLS model (Fig 1). Such insensitivity to the bootstrap analysis and accurate prediction of glacial *and* interglacial state $CO_2$ values would be unlikely in the case of major non-linear dependencies between the LR04 predictor and $CO_2$ response variables. Fourth, non-linear approaches would risk generating an improved fit due to statistical artefacts that do not meaningfully relate to any dependence between benthic $\delta^{18}O$ and $CO_2$. Finally, the specific causes and sources and sinks involved in glacial to interglacial and millennial-scale $CO_2$ variations still remain poorly constrained (e.g. Archer *et al*., 2000; Sigman *et al*., 2010; Gottschalk *et al*., 2019). Given this process-uncertainty, the GLS model fits our criteria of the simplest reasonable model. Further, the use of benthic $\delta^{18}O$ to predict atmospheric $CO_2$ has precedence; in response to the EPICA challenge (Wolff et al., 2004), N. Shackleton used this method to predict atmospheric $CO_2$ out to 800 kyr (Wolff, 2005). Furthermore, inverse modelling of $CO_2$ forced by the LR04 benthic stack has been undertaken by Berends et al. (2021a) and van de Wal et al. (2011)"

**Comment:** *"…discussing in more depth the underlying mechanisms relating benthic $\delta^{18}O$ and $CO_2$ to say more about the implications of potential misfit between $CO_2$ and the regression-based estimate."*

See lines 121-132 for discussion of mechanisms relating benthic $\delta^{18}O$ and $CO_2$:

"… Mechanistically, multiple processes are expected to contribute to the shared variance. A first order factor is the dependency of $CO_2$ solubility on ocean temperature (e.g. Millero, 1995). From the simple solubility perspective, colder climate states with increased ice volume and colder ocean temperatures will drive increased ocean uptake of $CO_2$ (Berends *et al.,* 2021). However, the solubility effect only accounts for a portion of observed glacial $CO_2$ drawdown (Archer *et al*., 2000). Multiple additional contributors to the shared variance are proposed in the literature. These include (not exhaustively), direct radiative forcing of ice volume changes by $CO_2$ (e.g. Shackleton *et al*., 1985); the impact of ice volume/sea level changes on

atmospheric $CO_2$ via ocean productivity and carbonate chemistry changes (e.g. Broecker, 1982; Archer *et al*., 2000; Ushie and Matsumoto, 2012); $CO_2$ drawdown during periods of high ice volume by increased iron fertilization (e.g. Röthlisberger *et al*., 2004; Martinez-Garcia *et al*., 2014) and enhanced sea ice extent during periods of high ice volume capping the ventilation of $CO_2$ from the ocean interior at high latitudes (Stephens and Keeling, 2000)."

**Comment:** *"Abstract, line 18: I think the authors meant benthic foraminiferal stable isotope (δ18O). The $\delta^{18}O$ data used is from foraminiferal calcite, not "water.""*

Revised to "LR04 benthic $\delta^{18}O_{calcite}$ stack (Lisiecki & Raymo, 2005) from marine sediment cores" at line 22 (abstract)

**Comment:** *"Line 118: It is not clear what the authors mean by "This trend is seen in our predicted record, and in the filtered BI-CO₂ and BOR-CO₂ data (Fig. 1C)." The previous sentence describes glacial stage $CO_2$ draw-down and the absence of an interglacial draw-down. In Fig. 1C, it appears that this description holds for the predicted $CO_2$ record (i.e., glacial draw-down but steady interglacial values). However, the BI-CO₂ and BOR-CO₂ data show a change in BOTH glacial and interglacial $CO_2$ compared to the post-MPT average. The text should be revised to make clear which trends are similar between the predictions and observations and which are different."*

In our revised manuscript we consider blue ice as the most reliable (currently available) measure of $CO_2$ concentration across the MPT for reasons noted in line 299-320 (see below). We quantitatively compare glacial and interglacial thresholds for blue ice data with our predictions and we no longer use BOR-CO₂ for these comparisons (see figure 3 – line 231). Our results section (lines 192-262) discusses glacial and interglacial trends exhibited by the two blue ice data sets (Yan *et al*, Higgins *et al*) through comparison to the current continuous records and we have worked to clarify the text about which trends are similar and which different between trends and observations:

> "We now consider long-term trends in interglacial and (separately) glacial $CO_2$ levels across the past 1.5 Myr in PRED-CO₂ and in the existing ice core $CO_2$ data. For PRED-CO₂ there is no significant difference between $CO_2$ concentrations in the interglacial stages of the 1.5 Mya ± 213 kya, 1000 ± 89 kya and 0–800 kya windows (Fig 4 D, blue bars). In the ice core observations, interglacial levels at 1.5 Mya in BI-CO₂ are also within the uncertainties of those in the 0–800 kya interval. Notably, the BI-CO₂ concentrations in the 1000 ± 89 kya interval appear elevated with respect to the 0–800 kyr and 1.5 Mya ± 213 kya intervals, however this elevated (ca. 271 ppm) level is consistent with the observed interglacial $CO_2$ concentration during interglacials 5, 9 and 11 (Fig 3B). Overall, there is no indication in the observed ice core $CO_2$ data or in PRED-CO₂ for a long-term trend in *interglacial* $CO_2$ levels across the past 1.5 Myr.
>
> In comparison, there are significant declines in glacial $CO_2$ levels across the MPT in PRED-CO₂ and the observed ice core data. For PRED-CO₂, glacial $CO_2$

concentrations are not significantly different during the 1.5 Mya ± 213 kya and 1000 ± 89 kya windows. However, across the MPT, PRED-$CO_2$ glacial concentrations drop by ~18 ppm. This pattern is consistent with the observed data, where glacial $CO_2$ levels are also not significantly different between the 1.5 Mya ± 213 kya and 1000 ± 89 kya windows (217.6 ± 2.3 and 226.2 ± 4.0 ppm, respectively) and then fall by 24 ppm to the 0–800 kyr observed glacial mean of 202.0 ± 3.2 ppm. Glacial-stage draw-down of $CO_2$ across the MPT in the absence of interglacial draw-down is consistent with previous observations based on the boron-isotope-based $CO_2$ reconstructions (e.g., Chalk *et al*., 2017; Hönisch *et al*., 2009 and see Discussion). In the following section we also compare PRED-$CO_2$ data to boron-isotope-based and other $CO_2$ proxy records covering the 0 to 1.5 Myr interval."

Lines 299-320 (Discussion) and Fig. 04 (line 337) quantitatively discuss boron-based $CO_2$ estimates in comparison to our predicted $CO_2$ record and their limitations:

"We consider the BI-$CO_2$ date to provide the most reliable measurements of $CO_2$ concentration, in the absence of a continuous ice core record across the MPT. However, further comparison of our $CO_2$ predictions can also be made against $CO_2$ proxy data from non-ice core archives (Fig 4A). We consider here $\delta^{11}$B-based atmospheric $CO_2$ reconstructions (Chalk *et al.,* 2017, Dyez *et al.* 2018 and Guillermic *et al.* 2022) and a recent atmospheric $CO_2$ reconstruction from $\delta^{13}$C of leaf wax (Yamamoto *et al.,* 2022). The continuous $\delta^{11}$B-based reconstructions of Dyez *et al.,* (2018) overlap PRED-$CO_2$ from ~1.38 – 1.5 Mya while the Chalk *et al.,* (2017) reconstruction overlaps PRED-$CO_2$ from 1.09 – 1.43 Mya. Discrete reconstructions from Guillermic *et al.* (2022) are distributed non-uniformly across the 800 to 1.5 Mya interval. For the two continuous $\delta^{11}$B-based reconstructions (Chalk *et al.,* (2017) and Dyez *et al.,* (2018)) the glacial $CO_2$ levels appear consistent with the PRED-$CO_2$ record, within their reported 30 – 60 ppm uncertainties. However, $\delta^{11}$B-based interglacial stages in these reconstructions exceed those of the PRED-$CO_2$ record (Fig. 4A). The Guillermic *et al.* (2022) reconstructions suggest a larger range of $CO_2$ concentrations than the overlapping intervals of PRED-$CO_2$ and of the two continuous $\delta^{11}$B-based reconstructions (Fig. 4A). The large range of the Guillermic *et al.* (2022) data and the high interglacial maxima in the Chalk *et al* (2017) and Dyez *et al.,* (2018) data, all significantly exceed the range and interglacial maxima from the BI-$CO_2$ estimates. These discrepancies internally between different $\delta^{11}$B-based $CO_2$ reconstructions and between the $\delta^{11}$B-based reconstructions and the BI-$CO_2$ data, may be due to uncertainties associated with the $\delta^{11}$B proxy transfer function. The $\delta^{11}$B-based $CO_2$ reconstructions are dependent on assumptions about multiple components of the carbonate system, including local marine carbon chemistry and the $CO_2$ saturation state in the past and (Hönisch *et al.,* 2009). Evidence that $\delta^{11}$B-based reconstructions may overestimate interglacial stage $CO_2$ is also seen in data from Chalk *et al.,* (2017) spanning ca. 0–250 kya, where the $\delta^{11}$B-based interglacial $CO_2$ levels exceed the continuous ice core $CO_2$ record by ca. 30 ppm (not shown)."

***Comment:*** *"Line 185-186: The authors need to explain why out-of-phase responses in northern and southern ice before the MPT (as proposed by Raymo et al., 2006) would lead them to expect "large discrepancies" between their regression-based $CO_2$ prediction and the*

*realized data. This inference seems to rely on the assumption of a certain relationship between $CO_2$ and northern or southern ice sheets, but I'm not sure what relationship the authors are assuming. Section 4.2 overall is quite short and would benefit from a more in-depth, process-based discussion of implications of the anti-phased hemisphere hypothesis for pre-MPT $CO_2$ variability"*

Lines 364-375 now offer a brief, but more in depth summary of the phase locking hypotheses:

> "The phase locking hypothesis is proposed to explain the absence of precession-related (23 kyr) periods in the LR04 benthic stack prior to the MPT (Fig 1), despite the strong precession cycle in insolation (Raymo *et al*., 2006, Morée *et al*., 2021). The key concept is that prior to the MPT the Northern Hemisphere and Antarctic ice sheets were responsive (in ice volume) to insolation changes in the precession band, but because precession forcing is out of phase between the hemispheres, the ice volume changes were opposing between the hemispheres and therefore cancelled in the benthic stack. This cancellation of the precession signal left insolation forcing in the 41 kyr obliquity band to dominate globally integrated ice volume changes expressed in the benthic stack. A transition from a smaller and more dynamic terrestrial-terminating Antarctic ice sheet to a larger and more stable marine-terminating ice sheet with cooling climate across the MPT (e.g. Elderfield *et al*., 2012) is then proposed to remove sensitivity of Antarctic ice volume to precession forcing and to suppress ice sheet sensitivity to the obliquity band in favour of quasi-100kyr ice volume changes that are in phase between the hemispheres (Raymo *et al*., 2006)."

We address/expand why out of phase responses in northern and southern hemisphere ice sheets would lead to large discrepancies between our model and a future realised $CO_2$ records from line 377-389:

> "Recently presented data from Yan *et al*. (2022), lend some support to the phase locking hypothesis, specifically with evidence that pre-MPT Antarctic temperature (and by extension ice volume) is positively correlated with a local precession-band insolation proxy based on the oxygen to nitrogen ratio of trapped air (Yan *et al*., 2022). Whereas the correlation becomes negative in the blue ice and continuous ice core data in the post-MPT record. If Yan *et al*., (2022) is correct and the phase locking hypothesis holds, then an implication is that prior to the MPT, Antarctic climate, Antarctic ice volume and by extension Southern Ocean climate conditions, would fall out of phase with the LR04 benthic stack. To now extend the argument to potential impacts on $CO_2$ exchange, if the phase locking hypothesis holds, then prior to the MPT the Antarctic and Southern Ocean climate conditions and by extension the Southern Ocean mechanisms of $CO_2$ exchange described earlier, would also be expected to fall out of phase with the benthic stack. Since our regression model assumes continuation of the in-phase relationship between the benthic stack and Antarctic and Southern Ocean climate conditions (as inherited from the post-MPT training data) we would expect to see major disagreement between our pre-MPT $CO_2$ predictions and a realised oldest ice continuous ice core $CO_2$ record."

**RC2:**

***Comment:*** *"line 16: "is to make""*

Accepted and revised

***Comment:*** *"line 17 and throughout the manuscript: Myr instead of myr"*

Accepted and revised

***Comment:*** *"line 25: the authors state that the null hypothesis should be rejected, however, without laying out the causal relationship between the regression parameters and potential reasons why the regression may not hold back in time, this statement is not entirely satisfying."*

We have undertaken further research and work for this significantly revised manuscript that has resulted in changes to our conclusions:

> 1: We have examined and include the more recently published blue ice core data at 1.5 Mya (Yan *et al*. (2022)) along with the blue ice data from 1 Mya (Higgins *et al.,* 2015).

> 2: To infer long term glacial and interglacial trends in observations compared to our predictions, we primarily focus on these two blue ice data sets.

> 3: We now suggest there is no clear evidence to reject our null-hypothesis.(line 399-406:

>> "We made initial tests of the null hypothesis by comparing our predicted $CO_2$ record to existing discrete blue ice $CO_2$ records and other non-ice-core proxy-$CO_2$ records from the 800–1500 kyr interval. Our predicted $CO_2$ concentrations do not show any systematic departure from observed blue ice $CO_2$ concentrations. The predictions are marginally lower (during glacial *and* interglacial stages) than those observed in blue ice from $1000 \pm 89$ kya and marginally higher than observed in blue ice data from $1.5$ Mya $\pm 213$ kyr. Our predictions were generally lower than interglacial $\delta^{11}$B-based-$CO_2$ reconstructions, but higher than recent $\delta^{13}$C of leaf-wax based $CO_2$ reconstructions. Overall, we do not find clear evidence from the existing blue ice or proxy $CO_2$ data to reject our predictions nor our associated null-hypothesis."

> 4: We have included more information on the relationship between the model parameters (See lines 117-132):

>> "Fig. 2 shows a scatter-plot of the LR04 $\delta^{18}$O benthic stack versus observed ice core $CO_2$ over the past 800 kyr. Both data sets are binned to equivalent 3-

kyr time steps (Methods). The Pearson's correlation coefficient (r) between the data sets is -0.82 (p < 0.05) indicating that ~68% of the variance in observed $CO_2$ is shared with the LR04 $\delta^{18}O$ benthic stack. This strong relationship provides an initial rationale for using the LR04 $\delta^{18}O$ benthic stack as an input parameter to predict $CO_2$ beyond 800 kyr. Mechanistically, multiple processes are expected to contribute to the shared variance. A first order factor is the dependency of $CO_2$ solubility on ocean temperature (e.g. Millero, 1995). From the simple solubility perspective, colder climate states with increased ice volume and colder ocean temperatures will drive increased ocean uptake of $CO_2$ (Berends et al., 2021). However, the solubility effect only accounts for a portion of observed glacial $CO_2$ drawdown (Archer et al., 2000). Multiple additional contributors to the shared variance are proposed in the literature. These include (not exhaustively), direct radiative forcing of ice volume changes by $CO_2$ (e.g. Shackleton et al., 1985); the impact of ice volume/sea level changes on atmospheric $CO_2$ via ocean productivity and carbonate chemistry changes (e.g. Broecker, 1982; Archer et al., 2000; Ushie and Matsumoto, 2012); $CO_2$ drawdown during periods of high ice volume by increased iron fertilization (e.g. Röthlisberger et al., 2004; Martinez-Garcia et al., 2014) and enhanced sea ice extent during periods of high ice volume capping the ventilation of $CO_2$ from the ocean interior at high latitudes (Stephens and Keeling, 2000)."

**Comment:** *"line 58-59: $\delta^{18}O$ is not just a sea level proxy but also influenced by deep ocean temperature. A process-based discussion of why LR04 is a viable input parameter to predict $CO_2$ is required."*

See lines 117-132 of the revised text (Introduction/above) for potential mechanistic relationships between $CO_2$ and $\delta^{18}O$, and lines 264-288 for a process based discussion of why LR04 is a viable input parameter to predict $CO_2$:

"Our objective with this manuscript was to generate the simplest reasonable model to predict $CO_2$ from the LR04 $\delta^{18}O$ benthic stack and to test the predictions against available observations. It is possible that the fit between observed and our predicted $CO_2$ data could be further improved using a non-linear approach. However, we refrain from a non-linear approach for several key reasons. First, a scatter plot of the LR04 $\delta^{18}O$ benthic stack versus observed ice core $CO_2$ over the past 800 kyr yields a Pearson's correlation coefficient (R) of -0.82 (Fig. 2), indicating that ~68% of the variance in observed $CO_2$ is shared with the benthic stack. Importantly, there is no evidence in this scatter plot for departure from the linear relationship at high or low $CO_2$ or benthic $\delta^{18}O$ levels. Second, following the approach of Chalk et al., 2017 and interpreting the upper 25th percentile of $CO_2$ data as representing mean interglacial stage $CO_2$ and the lower 25th percentile of $CO_2$ data as representing mean glacial stages $CO_2$ levels, we see that our predicted interglacial mean value for the past 800 kyr (253.1 ± 2.3 ppm) closely overlaps with the observed interglacial mean value (253.9 ± 4.1 ppm) and similarly, the predicted glacial stage mean (199.7 ± 1.7 ppm) closely overlaps with the observed glacial stage mean (202.0 ± 3.2 ppm). Third, the predictions are remarkably insensitive to bootstrap analysis in which 50 % of that data are omitted with each iteration of the GLS model (Fig 1). Such insensitivity to the bootstrap analysis and accurate prediction of glacial *and* interglacial state $CO_2$ values

would be unlikely in the case of major non-linear dependencies between the LR04 predictor and $CO_2$ response variables. Fourth, non-linear approaches would risk generating an improved fit due to statistical artefacts that do not meaningfully relate to any dependence between benthic $\delta^{18}O$ and $CO_2$. Finally, the specific causes and sources and sinks involved in glacial to interglacial and millennial-scale $CO_2$ variations still remain poorly constrained (e.g. Archer *et al.*, 2000; Sigman *et al.*, 2010; Gottschalk *et al.*, 2019). Given this process-uncertainty, the GLS model fits our criteria of the simplest reasonable model. Further, the use of benthic $\delta^{18}O$ to predict atmospheric $CO_2$ has precedence; in response to the EPICA challenge (Wolff et al., 2004), N. Shackleton used this method to predict atmospheric $CO_2$ out to 800 kyr (Wolff, 2005). Furthermore, inverse modelling of $CO_2$ forced by the LR04 benthic stack has been undertaken by Berends et al. (2021a) and van de Wal et al. (2011)."

**Comment:** *"Line 66: please include also the record by Dyez et al., Paleoceanography 2018"*

We have included the record presented by Dyez et al. in Fig. 04 (line 337) as a further boron-based $CO_2$ estimates. We discuss it in relation to our prediction about line 301-311:

"We consider here $\delta^{11}B$-based atmospheric $CO_2$ reconstructions (Chalk *et al.,* 2017, Dyez *et al.* 2018 and Guillermic *et al.* 2022) and a recent atmospheric $CO_2$ reconstruction from $\delta^{13}C$ of leaf wax (Yamamoto *et al.,* 2022). The continuous $\delta^{11}B$-based reconstructions of Dyez *et al.*, (2018) overlap PRED-$CO_2$ from ~1.38 – 1.5 Mya while the Chalk *et al.*, (2017) reconstruction overlaps PRED-$CO_2$ from 1.09 – 1.43 Mya. Discrete reconstructions from Guillermic *et al.* (2022) are distributed non-uniformly across the 800 to 1.5 Mya interval. For the two continuous $\delta^{11}B$-based reconstructions (Chalk *et al.,* (2017) and Dyez *et al.*, (2018)) the glacial $CO_2$ levels appear consistent with the PRED-$CO_2$ record, within their reported 30 – 60 ppm uncertainties. However, $\delta^{11}B$-based interglacial stages in these reconstructions exceed those of the PRED-$CO_2$ record (Fig. 4A). The Guillermic *et al.* (2022) reconstructions suggest a larger range of $CO_2$ concentrations than the overlapping intervals of PRED-$CO_2$ and of the two continuous $\delta^{11}B$-based reconstructions (Fig. 4A)."

and in relation to blue ice about line 311:

"The large range of the Guillermic *et al.* (2022) data and the high interglacial maxima in the Chalk *et al* (2017) and Dyez *et al.*, (2018) data, all significantly exceed the range and interglacial maxima from the BI-$CO_2$ estimates"

**Comment:** *"line 68: The very old ice at Allan Hills is not really from the surface but from a shallow ice drilling of more than 100 m depth."*

In this revised manuscript we review our initial description of blue ice about line 150-153:

"We use the term blue ice to describe deep, ancient glacial ice that has been brought nearer to the surface of an ice sheet by ice flow. Blue ice is sampled by cutting

trenches or shallow drilling of up to several hundred meters (e.g. Higgins *et al.,* 2015)"

**Comment:** *"Methods: the uncertainty in the regression connected to the independent age scales should be discussed"*

Within our methods section we discuss the age uncertainties associated with the LR04 benthic stack and the composite $CO_2$ record from line 178-190:

> "Uncertainties in the independent age scales of both the LR04 stack and the compiled $CO_2$ record are inherited by our GLS model and its predictions. The LR04 stack includes 57 globally-distributed benthic $\delta^{18}O$ sediment core records. The age models for these cores are independently constructed from the average sedimentation rates of each core, assuming global sedimentation rates have remained relatively stable, and with tuning to a simple ice model based on 21 June insolation at 65°N (Lisiecki & Raymo, 2005). The authors estimate uncertainty of 6 kyr from 1.5 – 1.0 Mya and 4 kyr from 1 – 0 Mya (Lisiecki & Raymo, 2005). The observed $CO_2$ composite ice core record for the past 800 kya (Bereiter at al., 2015) uses six independent dating methods for various core locations both spatially across Antarctica, and stratigraphically for different sections of the same core. The age uncertainty in the gas timescale has a median over the 0 – 800 kya interval of 2 kyr, but individual uncertainties can reach up to 5 kyr (Veres *et al* 2013; Bazin *et al*., 2013). The relative age uncertainties between these input variables may diminish the regression or in some instances lead to spurious correlation. However, we expect any such effects are minor on the basis that our predictions show little sensitivity to the bootstrap analysis with 1000 iterations of re-computing the regression after removing 50% of data (see Fig. 3B, C and Discussion)."

**Comment:** *"line 85: not clear what r(226) means, please explain. Did you allow for lag correlation? (see also comment on age scales above)"*

R(226) represented the degrees of freedom of the test to determine the correlation coefficient. For simplicity we have removed it. Further, Our generalised least square model accounted for autocorrelation/lag between the predictor ($\delta^{18}O$) and $CO_2$ using an AR(1) correlation factor.

**Comment:** *"line 89: the limitations of blue ice $CO_2$ reconstructions and $\delta^{11}B$ reconstructions of $CO_2$ should be discussed as well."*

Caveats associated with blue ice $CO_2$ measurements is now discussed at line 153-155 (Introduction):

"The vertical migration of blue ice is associated with high deformation making the ice samples stratigraphically complex and hard to date (Higgins *et al.,* 2015). As a result, blue ice records alone do not provide a continuous $CO_2$ record across the MPT."

and further at line 290-296 (Discussion):

"There are several caveats with blue ice data that may affect its use to evaluate our GLS model predictions. The blue ice data may have been subject to diffusional smoothing of $CO_2$ (e.g. Yan *et al.,* 2019), which would act in the direction of elevating the (lower 25[th] percentile) assumed glacial concentrations above the glacial atmospheric values and reducing the (upper 25[th] percentile) assumed interglacial concentrations. There is also the potential for artificially elevated $CO_2$ concentrations in blue ice due in-situ respiration of $CO_2$ due to microbial activity in detrital matter. Respiration effects are screened for by measurements of $\delta^{13}C$ of $CO_2$, however it is difficult to demonstrate that all samples are unaffected (Yan *et al*., 2019)."

Line 313-320 (Discussion) also covers limitations of $\delta^{11}B$ $CO_2$ reconstructions:

"These discrepancies internally between different $\delta^{11}B$-based $CO_2$ reconstructions and between the $\delta^{11}B$-based reconstructions and the BI-$CO_2$ data, may be due to uncertainties associated with the $\delta^{11}B$ proxy transfer function. The $\delta^{11}B$-based $CO_2$ reconstructions are dependent on assumptions about multiple components of the carbonate system, including local marine carbon chemistry and the $CO_2$ saturation state in the past and (Hönisch *et al*., 2009). Evidence that $\delta^{11}B$-based reconstructions may overestimate interglacial stage $CO_2$ is also seen in data from Chalk *et al*., (2017) spanning ca. 0–250 kya, where the $\delta^{11}B$-based interglacial $CO_2$ levels exceed the continuous ice core $CO_2$ record by ca. 30 ppm (not shown)."

**Peter Kohler:**

**Comment:** *"To be transparent in what has been done, the equation which calculates $CO_2$ out of the LR04 benthic $\delta^{18}O$ stack is missing. Plotting of the LR04 benthic $\delta^{18}O$, which is at the core of the approach is also missing."*

Line 171 now shows the equation used to calculate $CO_2$ from $\delta^{18}O$, and the LR04 benthic plot has been added to figure 3a (line 231).

**Comment:** *"Blue ice $CO_2$ data from Allan Hills have been extended in Yan et al (2019), now also containing snapshots of $CO_2$ at 1.5 and 2.0 Ma"*

Thank you. We have included the blue ice data at 1.5 Mya by Yan *et al.* (see figure 3 – line 231). In the manuscript lines 206-262 (Results) we compare both the Higgins *et al.,* and Yan *et al.,* blue ice data to each other, to the current 800 kya record and to our predictions: See manuscript lines 192 to 262:

"We now consider long-term trends in interglacial and (separately) glacial $CO_2$ levels across the past 1.5 Myr in PRED-$CO_2$ and in the existing ice core $CO_2$ data. For PRED-$CO_2$ there is no significant difference between $CO_2$ concentrations in the interglacial stages of the 1.5 Mya ± 213 kya, 1000 ± 89 kya and 0–800 kya windows (Fig 4 D, blue bars). In the ice core observations, interglacial levels at 1.5 Mya in BI-$CO_2$ are also within the uncertainties of those in the 0–800 kya interval. Notably, the BI-$CO_2$ concentrations in the 1000 ± 89 kya interval appear elevated with respect to the 0–800 kyr and 1.5 Mya ± 213 kya intervals, however this elevated (ca. 271 ppm) level is consistent with the observed interglacial $CO_2$ concentration during interglacials 5, 9 and 11 (Fig 3B). Overall, there is no indication in the observed ice core $CO_2$ data or in PRED-$CO_2$ for a long-term trend in *interglacial* $CO_2$ levels across the past 1.5 Myr.

In comparison, there are significant declines in glacial $CO_2$ levels across the MPT in PRED-$CO_2$ and the observed ice core data. For PRED-$CO_2$, glacial $CO_2$ concentrations are not significantly different during the 1.5 Mya ± 213 kya and 1000 ± 89 kya windows. However, across the MPT, PRED-$CO_2$ glacial concentrations drop by ~18 ppm. This pattern is consistent with the observed data, where glacial $CO_2$ levels are also not significantly different between the 1.5 Mya ± 213 kya and 1000 ± 89 kya windows (217.6 ± 2.3 and 226.2 ± 4.0 ppm, respectively) and then fall by 24 ppm to the 0–800 kyr observed glacial mean of 202.0 ± 3.2 ppm. Glacial-stage draw-down of $CO_2$ across the MPT in the absence of interglacial draw-down is consistent with previous observations based on the boron-isotope-based $CO_2$ reconstructions (e.g., Chalk *et al*., 2017; Hönisch *et al*., 2009 and see Discussion). In the following section we also compare PRED-$CO_2$ data to boron-isotope-based and other $CO_2$ proxy records covering the 0 to 1.5 Myr interval."

***Comment:*** *"A recent paper by Yamamoto et al (2022) calculates $CO_2$ over the MPT from leaf wax $\delta^{13}C$ and finds that smaller glacial/interglacial amplitudes in $CO_2$ before the MPT are based on stable glacial $CO_2$, but smaller interglacial $CO_2$ before the MPT. This differs to the $\delta^{11}B$-based $CO_2$, and if I got it right might support the here defined Null Hypothesis, which then cannot easily be dismissed."*

Another great recommendation for data to examine. This has been included in our discussion sections in figure 4A (line 337). We discuss it specifically at line 322-325:

"By comparison, the $\delta^{13}C$ of leaf wax data (Yamamoto *et al.,* 2022) has a similar glacial to interglacial range as PRED-$CO_2$, but a ca. 20ppm lower mean concentration than our predictions (Fig 4A). Hence, our PRED-$CO_2$ data fall lower than interglacial $\delta^{11}B$-based interglacial levels but are higher than the $\delta^{13}C$ of leaf-wax based estimate. Given the evidence that $\delta^{11}B$-based reconstructions are known to overestimate atmospheric $CO_2$ concentration in the continuous ice core record, we do not find cause from the existing $CO_2$ proxy data to reject our predictions nor our associated null-hypothesis."

**Comment:** *"New $CO_2$ data based on $\delta^{11}B$ from Pacific cores have recently been published (Guillermic et al., 2022). Ok, data coverage across the last 1.5Ma might be weak, but worth discussing it."*

Boron based $CO_2$ estimates by Guillermic *et al.,* have been added to figure 4A (line 337) and discussed with other boron based estimated at line 301-320:

> "We consider here $\delta^{11}B$-based atmospheric $CO_2$ reconstructions (Chalk *et al.,* 2017, Dyez *et al.* 2018 and Guillermic *et al.* 2022) and a recent atmospheric $CO_2$ reconstruction from $\delta^{13}C$ of leaf wax (Yamamoto *et al.,* 2022). The continuous $\delta^{11}B$-based reconstructions of Dyez *et al.,* (2018) overlap PRED-$CO_2$ from ~1.38 – 1.5 Mya while the Chalk *et al.,* (2017) reconstruction overlaps PRED-$CO_2$ from 1.09 – 1.43 Mya. Discrete reconstructions from Guillermic *et al.* (2022) are distributed non-uniformly across the 800 to 1.5 Mya interval. For the two continuous $\delta^{11}B$-based reconstructions (Chalk *et al.,* (2017) and Dyez *et al.,* (2018)) the glacial $CO_2$ levels appear consistent with the PRED-$CO_2$ record, within their reported 30 – 60 ppm uncertainties. However, $\delta^{11}B$-based interglacial stages in these reconstructions exceed those of the PRED-$CO_2$ record (Fig. 4A). The Guillermic *et al.* (2022) reconstructions suggest a larger range of $CO_2$ concentrations than the overlapping intervals of PRED-$CO_2$ and of the two continuous $\delta^{11}B$-based reconstructions (Fig. 4A). The large range of the Guillermic *et al.* (2022) data and the high interglacial maxima in the Chalk *et al* (2017) and Dyez *et al.,* (2018) data, all significantly exceed the range and interglacial maxima from the BI-$CO_2$ estimates. These discrepancies internally between different $\delta^{11}B$-based $CO_2$ reconstructions and between the $\delta^{11}B$-based reconstructions and the BI-$CO_2$ data, may be due to uncertainties associated with the $\delta^{11}B$ proxy transfer function. The $\delta^{11}B$-based $CO_2$ reconstructions are dependent on assumptions about multiple components of the carbonate system, including local marine carbon chemistry and the $CO_2$ saturation state in the past and (Hönisch *et al.,* 2009). Evidence that $\delta^{11}B$-based reconstructions may overestimate interglacial stage $CO_2$ is also seen in data from Chalk *et al.,* (2017) spanning ca. 0–250 kya, where the $\delta^{11}B$-based interglacial $CO_2$ levels exceed the continuous ice core $CO_2$ record by ca. 30 ppm (not shown)."

**Comment:** *"$CO_2$ as function of benthic $\delta^{18}O$ has in an inverse modelling approach already been calculated by Stap et al (2016). This approach has been updated by Berends et al. (2021a). So comparison to their results might tell, how (if at all) this study shows something new."*

Our $CO_2$ prediction, like Berends et., 2021a, was trained on data from the recent 800 kyr and motivated by comparison to the upcoming oldest ice core records. Our simple model yields a high correlation to the observed 800 kyr Bereiter et al., $CO_2$ record ($r^2 = 0.68$) and our $CO_2$ predictions out to 1.5 Myr can not be confidently excluded by the available blue ice and $CO_2$ proxy reconstructions. From what we understand Berends et., 2021a does not make any evaluation or comparison to the discrete $\delta^{11}B$ and blue ice data over the MPT.

**Comment:** *"Maybe also discuss other approaches of CO₂ across the MPT, eg C cycle simulation results (apart from those in Willeit et al, 2020, which are cited) of Köhler & Bintanja (2006), or the compilation of at that time available CO₂ data and the calculation of a continous high-resolution CO₂ record in van de Wal et al. (2011), updated in Stap et al. (2018)."*

Figure 4B (line 337) now includes the modelled record by van de Wal (*et al*) and Willeit (*et al*) and these are discussed, relative to our predictions at line 329-335:

> "We also compare our predictions to existing more complex model simulations (Fig 4B.). First, against a transient simulation using an intermediate-complexity earth system model (CLIMBER-2) by Willeit *et al.* (2019). This study suggests a combination of gradual regolith removal and atmospheric CO₂ decline can explain the long-term climate variability over the past 3Myr. Second, against a longer-term reconstruction by van de Wal *et al.* (2011) that utilises deep-sea benthic isotope records to reconstruct a continuous CO₂ record over the past 20 Myr. Our simple GLS model demonstrates a similar long-term trend and timing of glacial-interglacial signals and an atmospheric CO₂ level that sits approximately mid-way between the two more complex models."

**Comment:** *"The recent review on the MPT (Berends et al., 2021b) gives also an idea about processes including a collection of CO₂ data and discusses a potential influence of the carbon cycle on the climate transition."*

We have included a reference to this excellent review (line 82) and have also referenced it throughout the paper.

**Comment:** *"While mentioning the call for the EPICA challenge, maybe also cite / discuss its results (Wolff et al., 2005). They have been shown on 2 posters at AGU fall meeting in 2004 (PDFs for download at: https://epic.awi.de/id/eprint/11721/, https://epic.awi.de/id/eprint/11722/), on which you see, that one of the participants to the challenge (N Shackleton) also used δ¹⁸O to predict CO₂ for the 400-800 ky time window."*

We refer to the results of the EPICA challenge (particularly the precedence set by N Shackleton) at line 285 of the manuscript:

> "… the use of benthic δ¹⁸O to predict atmospheric CO₂ has precedence; in response to the EPICA challenge (Wolff et al., 2004), N. Shackleton used this method to predict atmospheric CO₂ out to 800 kyr (Wolff, 2005). Furthermore, inverse modelling of CO₂ forced by the LR04 benthic stack has been undertaken by Berends et al. (2021a) and van de Wal et al. (2011)."

---

## Author Response (AR2)

Thank you Constantijn for the detailed feedback and comments on the manuscript. We respond to all comments below and include a tracked-changes version of the manuscript showing the changes to the main text and figures that we have made in response. The review questions/comments are in blue and our response in black. Line number references refer to the tracked changes version of the manuscript. In the revised text we have also made some minor formatting, spelling and small text adjustments to improve readability.

*Now that you show the details of the d18O-CO2 regression more clearly, and the improved comparison to CO2 records, the results of this study are near-identical to the work of Berends et al. (2021), which also included a statistical model for prediction. To ensure that this new work has novelty, please highlight in the main text whether this work extends or replicates Berends et al. (2021) and explain how the approaches may have been different, or not.*

Thanks for this comment. We agree it is important to better clarify the differences between our approach and Berends et al., 2021. We make three main points in response and show further below the changes made to the main text:

1. Berends et al., 2021 uses an inverse forward modelling approach, which is quite different and more complex than our GLS approach. We add text to the introduction, as shown further below, which references Berends et al. (and the preceding van de Wal 2011 and Stap et al., 2016 studies) and gives more context on the different assumptions in the inverse modelling approach compared to our GLS approach.

2. Thanks for pointing out that Berends et al also included a statistical (ordinary least squares regression) prediction of $CO_2$ from the benthic stack with $r^2 = 0.7$ as a point of comparison to the main result in that paper which is the inverse model. To be fair, the statistical model is a minor part of their paper and is not plotted in the paper nor provided in the data supplement. Our statistical model uses generalised least squares (GLS), rather than ordinary least-squares (OLS). While both models assume Linearity between variables, GLS allows heteroskedasticity in the variances and autocorrelation in the error terms. We used Autocorrelation Function (ACF), and Partial Autocorrelation Functions (pACF) and determined autocorrelation was present between observations and therefore that OLS is not reliable for parameter estimation. In the main text we expand on the GLS and AR(1) methodology used on our paper.

3. Our GLS reconstruction is compared to proxy data not available at the time of the Berends et al. study. Including recent reconstructions from boron data (Guillermic et al., 2022) and leaf wax $\delta^{13}C$ (Yamamoto et al., 2022). We also include comparison of our modelled $CO_2$ record to two sets of blue ice $CO_2$ data (Higgins et al., 2015; Yan et al., 2022), whereas the Berends et al. study does not make any comparisons with blue ice $CO_2$. The blue ice comparisons are particularly important as they are $CO_2$ measurements rather than proxy reconstructions and we calculate and compare mean glacial and interglacial concentrations between our predictions and blue ice records (Fig 3b, c.), which is a novel way of handling the dating uncertainty in the blue ice data.

**Changes to the main text in response to this comment:**

In the Introduction we add text to clarify the difference between our statistical GLS approach and the inverse modelling approach in Berends et al. 2021. New text in italics from Line 149:

> .. we make the simple assumption that the relationships between the LR04 benthic $\delta^{18}O$ stack and $CO_2$ can be extended beyond 800 kya and use generalised least squares (GLS) regression modelling between benthic $\delta^{18}O$ and $CO_2$ to make a prediction of $CO_2$ spanning 800–1500 kya. The deliberately simple implicit assumption, and null hypothesis, is that there is no change to the feedback processes linking benthic $\delta^{18}O$ and $CO_2$ before and after the MPT.

> *[Our] approach differs to previous more complex model studies that have attempted to reconstruct $CO_2$ using the LR04 benthic $\delta^{18}O$ stack as an input variable (van de Wal, 2011; Stap et al., 2016, Berends et al., 2021b). The latter studies use an inverse forward modelling approach, in which climate and ice sheet models of various complexities are used to capture physical relations between $CO_2$, global temperature and ice volume. For example, in Berends et al., 2021b the offset between modelled and observed benthic $\delta^{18}O$ is used to calculate a value for atmospheric $CO_2$ that is iterated back to the inverse model. The $CO_2$ record which minimises the difference between the modelled and observed benthic stack is then taken as an estimate of how atmospheric $CO_2$ may have evolved to force coupled climate, deep ocean temperature and land ice volume changes that reproduce the observed benthic $\delta^{18}O$ signal. Accuracy of the reconstructions in the inverse modelling approach depends on the ability of the climate and ice sheet models used to capture the correct climate dynamics across the MPT. Our GLS method is a simpler statistical approach, designed with the specific null hypothesis in mind, that does not attempt to simulate the physics linking benthic $\delta18O$ signal, land ice volume, global temperature and $CO_2$. A range of approaches to reconstructing $CO_2$ have been called for and are of value in the context of forthcoming continuous ice core records across the MPT from oldest ice projects currently underway in Antarctica [IPICS 2020].*

We also add the Berends et al., 2021 inverse model reconstruction of $CO_2$ to our Fig. 4b. And add some text to the Discussion comparing the results, which despite the different approaches do lead to quite similar reconstructions.

We also note the inclusion of the OLS approach within Berends et al., 2021 around Line 308:

> *"This is similar to that reported in ordinary Linear least-squares regression ($r^2$=0.70) by Berends et al. (2021b)".*

*Why did you use the LR04 benthic stack rather than the more recent stack by Ahn et al. (2017)? (**Line 11**)*

Differences between Ahn et al., (2017) and the LR04 benthic stack are small for the past 1.5 Myr. We decided to use LR04 because and for consistency with previous model studies which use that version of the benthic stack as an input variable in reconstructing $CO_2$, e.g. van de Wal et al. (2011), Stap et al. (2016), Berends et al. (2021b).

*Lines 73-74: "Emergence of … after the MPT (Fig. 1A)" While the spectral power in Fig. 1 does indeed show a peak around 100 kyr, which coincides with the eccentricity cycle, this kind of analysis is problematic. The "skipped obliquity cycle hypothesis" posits that the post-MPT glacial cycles are mostly alternating 80/120-kyr. Such a signal shows up in a spectral analysis as a single 100-kyr peak, not as separate 80 and 120-kyr peaks (try it out if you like!), so that this result by itself is not conclusive. Please discuss this.*

Good point. We add the following text to the main text at Line 60:

> *Indeed, the skipped obliquity cycle hypothesis, proposes that the 100 kyr signal seen in spectral analysis of the post-MPT global benthic $\delta^{18}O$ stack (e.g. Fig 1A) may be comprised of alternating 80 and 120-kyr signals, i.e. in which the intervening obliquity cycles are skipped.*

And further down (e.g. Line 76) we refer to the '100 kyr signal' rather than 'eccentricity signal'.

*Lines 113-115: please add some additional text regarding the combined record of bottom water temperature and global ice volume. For the 0-800 ka time interval the proxy record of Elderfield et al. (2012) showed that the shape of the glacial-interglacial changes are quite different between these two components, which may also impact your premise that Southern Ocean is an important link. It's not a Linear relationship between ice volume and ocean temperature.*

We add additional text to the main text as below (in italics) in response at Line 118, we clearly note the combined signals and the non-Linear relationship.

> The $\delta^{18}O$ ratios in the LR04 benthic stack are governed primarily by deep ocean temperature and global ice volume at the time the foraminifera lived, with higher values indicating both increased ice volume and a colder climate. *The relationship between the ice volume and ocean temperature components contributing to the $\delta^{18}O$ benthic stack are not Linear. Separating the two signals remains challenging and has been attempted elsewhere using a range of approaches from comparison with paired deep ocean temperature proxies (Elderfield et al., 2012), inverse modelling (Berends et al., 2021b) and spectral analysis (e.g. Huybers and Wunsch, 2009).*

> As explained in the main text (see from Line 141), our paper does not attempt a quantitative separation and attribution of the processes linking global ice volume, ocean temperature and atmospheric $CO_2$. In this way our statistical approach is different to inverse modelling or other modelling approaches that attempt to represent the complete physics. We note clearly that our approach limits us to testing a simple implicit null hypothesis through comparison with existing and forthcoming data.

*Line 158 presents the reconstruction by van de Wal et al. (2011) but the more recent version by Berends et al. (2021) should be presented since it has updated ice sheet and climate models.*

We have added the Berends et al., (2021b) inverse model $CO_2$ reconstruction to Fig. 4B and some text describing the model in the Discussion (ca. Line 370-376) as follows:

> *Our simple GLS model demonstrates a similar long-term trend and timing of glacial-interglacial signals and an atmospheric $CO_2$ level that sits approximately mid-way between*

*the van de Wal et al. (2011), and Willeit et al. (2019) models and is remarkably similar to the Berends et al., 2021b reconstruction despite their different approaches. Notably the Berends et al. reconstruction shows greater glacial to interglacial amplitude in the $CO_2$ signal compared to our GLS-model. The decreasing Linear trend in CO2 in Willeit et al. (2019), which is not seen in the other reconstructions, was directly prescribed in that study to induce Northern Hemisphere glaciation at 2.6 Myr ago.*

Please refer also to the abovementioned additions to the Introduction and Discussion citing Berends et al. 2021b and the inverse model approach.

*Line 171: there needs to be some additional text to explain what this "autoregressive factor" represents.*

We revised the text around Line 189 to clarify our use of both the GLS technique and the AR(1) factor:

*We use a generalised least squares (GLS) model with an auto-regressive (AR) factor 1 to predict atmospheric $CO_2$ from the LR04 benthic $\delta^{18}O$ stack (Fig. 3A and B). We use GLS because the assumptions of ordinary least squares (OLS) are violated by the presence of autocorrelation and heteroskedasticity in the regression errors. We selected the AR(1) correlation factor as it yielded the lowest Akaike information criterion (AIC) value from a test of multiple correlation factors. The AR(1) process assumes and accounts for dependence of error at a given point in time on the previous error term. In practise this makes the model assumptions more realistic and improves parameter estimation where, as in the climate system, observations are dependent on past values.*

*Line 190: "after removing 50% of data". Please provide additional explanation on how this works, and what the results are. If your data is significantly auto-correlated (which with a 3-kyr time step I think it is) then I'd think it's not very surprising that removing some of the data points (which essentially don't contain any new information) doesn't affect the least-squares fit.*

Each iteration of the model removes a different, random 50% of the data. Doing this with replacement introduces variability among the bootstrap samples. Since each sample can have different combinations of the original data points (including repeated ones), this variability helps in assessing the robustness and stability of the model. We clarify the text around Line 205. We don't expect the bootstrap method to address auto-correlation but is an accepted method to gauge sensitivity to data and dating uncertainty.

*To gauge the GLS model stability we took a bootstrap approach, selecting a random 50% subset of our data (with replacement) and re-running the model 1000 times to determine 95% confidence intervals for the predictions. While the GLS method itself addresses autocorrelation, the bootstrap method introduces variability such that each iteration of the model has different combinations of the original data points (including repeated ones), this variability helps in assessing the robustness and sensitivity of the model, e.g. to variable data and dating uncertainty.*

*Lines 261-262: "In the following section … interval" Why do you compare to some data in the Results section, and then to some other data in the Discussion section?*

We decided to split into results and discussion accordingly:

> Results: Comparison of our $CO_2$ record to those data that contain measured atmospheric concentrations of $CO_2$, specifically blue ice data from Higgins et al. and Yan et al.

> Discussion: *Proxy (e.g. boron and leaf wax)* and model reconstructions of atmospheric CO2.

This separation is to distinguish between ice core data, which is an actual measurement of past atmospheric concentration, versus proxy or model data, which are reconstructions that are based on a range of assumptions. Once can see by the spread between overlapping proxy and model data series that the records are inconsistent with each other and therefore the proxy and model data cannot alone be used to refute our predictions (we note a similar point is made in Berends et al., 2021b). So we think these comparisons to proxy and model are suited to the Discussion. A weakness of the blue ice data is the dating uncertainty, which we address quantitatively in the Results by calculation of mean glacial and interglacial ranges for the overlapping GLS predictions and blue ice measurements.

We add a Line to the main text to make this approach more clear to the reader (Line 179):

> *In the Discussion, we also compare our predicted record to existing proxy-$CO_2$ reconstructions…*

*Line 325: this statement is not supported by a citation, please include reference to the evidence that boron isotopes over-estimate CO2.*

We have modified this statement, it no longer refers specifically to δ11B-based reconstructions specifically over-estimating $CO_2$ concentration. Instead we refer to the large spread in existing proxy-$CO_2$ reconstructions, as follows (Line 358).

> "*The strong spread between these different proxies and the large associated uncertainty of the alternative marine and leaf wax proxy-$CO_2$ reconstructions mean that we do not find cause from the existing $CO_2$ proxy data to reject our predictions nor our associated null-hypothesis.*"

For information though, we do see evidence for overestimation by $\delta^{11}B$-based estimates in data from Chalk et al., 2019 overlapping the observed continuous ice core record as shown below. We do not include this in the main text since on reflection, we do not have evidence that this overestimation is systematic for pre-MPT data or other $\delta^{11}B$-based reconstructions.

[Figure]

Fig R1. The continuous ice core composite atmospheric $CO_2$ record in orange (Bereiter at al., 2015) and reconstructed atmospheric $CO_2$ from boron isotopes for the interval 0–250k in blue (Chalk et al., 2017).

*Lines 350-351: clarify here whether you are referring to the whole time interval shown on your figures (0-1.5 Ma), or a narrower period? Figure 3 seems to suggest that glacial CO2 was fairly constant >1 Ma, then declined through ~1000 to ~650 ka. The text currently doesn't distinguish between these scenarios, which have quite different implications for climate forcing.*

Agreed, this is an important point to clarify. Here we refer to figure 3b, we see a drop in glacial CO2 across the MPT when comparing the regions before and after the grey shaded region (representing the MPT as defined by Chalk et al. (2017)).

We adjust the main text accordingly at Line 396:

> *That fact that our LR04-based prediction of $CO_2$ captures this same trend, with predicted glacial $CO_2$ fairly constant from 1.5 to ca. 1.0 Mya before declining from 1.0 to 0.6 kya, reflects that the LR04 benthic stack features an increase in the interglacial to glacial benthic $\delta^{18}O$ difference across this same interval, which is dominated by the glacial stage changes (Fig 3A.).*

---

## Author Response (AR3)

Review response – Lorraine Lisiecki

Dear Erin,

The series of reviews have greatly strengthened this manuscript, and we appreciate the additional constructive comments from Lorraine Lisiecki. The requested corrections/clarifications to references to previous work along with all other suggestions have been taken up in full. Please refer to the line-by-line response below to each review point and to the tracked changes version of the manuscript attached to see these implemented changes.

In response to comment 5. from this review we have extended the model run from 1.5 Myr to 1.8 Myr. This allows us to make a comparison between corresponding time windows for PRED-CO2 and the full dating uncertainty of the Yan et al., 2019 blue ice data. This extension will also enhance the value of the PRED-CO2 data set for comparison with upcoming oldest ice records, which new geophysical data is indicating may extend to 1.8 Myr.

All code has been reviewed and prepared for upload as has the PRED-CO$_2$ data set. In reviewing the code we identified a couple of rounding errors and typos that have a minor effect on some quoted numbers in the text. Importantly, none of these changes affect the significance of comparisons between observed and predicted data or any conclusions. The corrected numbers are all noted in the tracked changes version and are summarised in a table at the end of the review response for full clarity.

Line numbers refer to the tracked changes version of the manuscript.

Sincere thanks for the extensions received to address earlier reviews while the authors managed other commitments.

Best regards, Jordan, Joel and Tess

**1. Line 81: Use (Raymo et al., 2006) citation throughout text for hypothesis #3. Raymo & Huybers (2008) was a short review of several existing hypotheses. It didn't present any new details about the antiphase hypothesis.**

Accepted and revised @ line 74-78

> *"Emergence of significant precession and 100 kyr signals occurs across the MPT (Fig. 1B), and all three components are clearly present after the MPT (Fig. 1A). Raymo et al. (2006) suggested that precession-paced changes in northern and southern hemisphere ice volumes may have occurred prior to the MPT, but are cancelled due to out-of-phase ice volume changes between the two hemispheres"*

**2. Line 97: Whether a decrease in CO2 is expected during both glacials and interglacials depends on assumptions about the causes and effects of CO2 change as demonstrated by the authors' later explanation of why CO2 might decrease only during glacial stages (lines 99-105). Therefore, I recommend changing the wording here to say "reduction in CO2 might be expected in both" or "reduction in CO2 has been proposed in both…"**

Accepted and revised @ line 93-94:

*"For a long-term decrease in radiative forcing by atmospheric CO2 to be the cause of the MPT, the reduction in CO2 might be expected in both glacial and interglacial stages…"*

**3. Line 216: This description of the LR04 age model development is unclear because it suggests that each individual core's age model was based on that core's sedimentation rate. I recommend revising it to "The age models for these cores are constructed by alignment of their d18O signals, followed by tuning of the stack to a simple ice model based on 21 June insolation at 65°N in a way which maintains relatively stable global mean sedimentation rates."**

Accepted and revised @ line 211-213:

> *"The LR04 stack includes 57 globally-distributed benthic δ18O sediment core records. The age models for these cores are constructed by alignment of their δ18O signals, followed by tuning of the stack to a simple ice model based on 21 June insolation at 65°N in a way which maintains relatively stable global mean sedimentation rates."*

**4. Line 225: Please clarify the meaning of "(median, 2 sigma, 5.78 ppm)." I think this is supposed to indicate that the 2-sigma value for all bootstrap analyses has a median of 5.78 ppm, but it isn't clear from the current notation.**

Strictly, what we get from the 1000 iterations of the model is a 95% confidence interval in the uncertainty of PRED_CO2 at each timestep. These uncertainties are provided in the PRED-CO2 data file. The median of the bootstrap 95% CI from 0 to 1.8 Mya is 5.78 ppm. We rephrase to (line 222-224)

> *"However, we expect any such effects are minor on the basis that our predictions show little sensitivity to the bootstrap analysis, which has a median 95% confidence interval of 5.8 ppm from 0 to 1.8 Mya (see Fig. 3B, C and Discussion)."*

**5. Line 243-244 (and throughout the results section): It's not clear to me how the blue ice time window of 1.5 Mya +/- 213 kya is compared to the PRED-CO2 values which appear to only extend to 1.5 Mya. Do the authors only use the PRED-CO2 values from 1.287 to 1.5 Mya? That wouldn't be a completely fair comparison because the blue ice may be affected by higher or lower atmospheric CO2 levels during the older half of the time window (1.5 to 1.713 Mya). The authors should clarify whether they analyze PRED-CO2 older than 1.5 Mya without showing it in their figures or use different time windows for calculating the average CO2 values from the blue ice and predicted CO2. If they use different time windows, the text should include a caveat that this presents a potential source of bias in the comparison.**

This is a good point. It is correct that the comparison to PREDCO2 is up to 1.5 Myr, whereas the Yan et al., blue ice data has an uncertainty range spanning 1.287 to 1.713 Mya. This reflects legacy of the Yan data being introduced for comparison during the review process, without updating the length of the model run. Note that this is not a time range covered by the blue ice data but the dating uncertainty range of the data.

To remove any concern about a potential source of bias and fully address this point, we have now extended the model prediction to 1.8 Myr and the extended PRED-$CO_2$ time series is now shown in Fig. 2 and 3.

The model extension allows a direct comparison of the Yan et al., and PRED-$CO_2$ data over the identical 1.5 Mya +/- 213 kya window. The impact on the comparison on the mean values is small <4 ppm. For clarity we show below (Table R1) the previous values and the values after extending PRED-$CO_2$. Notably the extension of the time widow (more data points) lowers the standard error in the uncertainty bound for PRED-$CO_2$. There remains no difference between the interglacial BI-$CO_2$ and interglacial PRED-$CO_2$ over the extended time window (as seen in Fig 3D). The central values for the glacial BI-CO2 and PRED-CO2 barely change (<1 ppm), but the reduced uncertainty bound translates to a 2.9 ppm difference between the upper estimate of BI-$CO_2$ and the lower estimate of PRED-$CO_2$ (see R1 below). We regard this difference as marginal, particularly given caveats associated with the blue ice dating and concentrations, that are covered in the Discussion. We update the text to reflect the extension of the time interval and revised comparison as follows (Line 265-273):

> *During the 1.5 Mya ± 213 kyr interval, the mean BI-$CO_2$ concentration did not show any significant difference to PRED-$CO_2$ in interglacial stages (254.1 ± 10.3 versus the predicted 257.2 ± 1.7 ppm. During glacial stages there is a small (2.9 ppm) difference between the upper estimate of BI-$CO_2$ and the lower estimate of PRED-$CO_2$ (218.4 ± 1.3 and 224 ± 1.4 ppm respectively, see Fig 3D). In our view these results, notwithstanding the 2.9 ppm difference at 1.5 Mya, do not give any cause to reject the GLS model. Furthermore, the comparison indicates that PRED-$CO_2$ is not drifting systematically away from the existing observed BI-$CO_2$ data (Fig 3D). The differences could of course be a failing in the model, potential biases in the blue ice data, dating uncertainty and/or unconstrained uncertainties (see Discussion for blue ice caveats).*

Throughout the manuscript references to a 1.5 Myr prediction of $CO_2$ are changed to 1.8 Myr.

**Table R1 changes in numbers due to the extension of the model from 1.5 Mya to 1.8 Mya**

| Line | Data | Stage | i.e. | Period | Original | New | Reason |
|---|---|---|---|---|---|---|---|
| 268/303 | Yan blue ice | Glacial | Fig 3d | 1.5 Mya ± 213 kyr | 217.6 ± 2.3 | 218.4 ± 1.3 | Review |
| 267 | Yan blue ice | Interglacial | Fig 3d | 1.5 Mya ± 213 kyr | 256.3 ± 3.8 | 254.1 ± 10.3 | Review |
| 268 | Predictions | Glacial | Fig 3d | 1.5 Mya ± 213 kyr | 224.2 ± 6.6 | 224 ± 1.4 | Model ext. |
| 267 | Predictions | Interglacial | Fig 3d | 1.5 Mya ± 213 kyr | 261.1 ± 6.3 | 257.2 ± 1.7 | Model ext. |

**6. Line 295: How do the authors conclude that glacial CO2 levels of 217.6 +/- 2.3 and 226.2 +/- 4.0 ppm are NOT significantly different from each other? The 95% confidence intervals for the two estimates do not overlap (if the uncertainties quoted are 95% confidence intervals as described on line 240). Perhaps there is a typo here?**

Thank you for catching this. The difference is small but significant at the quoted CI. However, this is in the direction of a marginal *increase* in $CO_2$ between the 1.5 Mya ± 213 kya (Yan et al., 2019) and 1000 ± 89 kya (Higgins et al., 2015) blue ice data sets. The main point the manuscript makes here is that there is not a decline in glacial $CO_2$ with time between these intervals, wheareas there is a very clear (ca. 24ppm) decline in glacial stage $CO_2$ between the 1.0 Myr and 0 – 800 kyr interval that follows (see Fig 3D). We adjust as follows. (Line 301-304):

> *"This pattern is similar to the observed BI-CO$_2$ data, where glacial CO$_2$ levels show no decline between the 1.5 Mya ± 213 kya and 1000 ± 89 kya windows (indeed there is a marginal increase from 218.4 ± 2.3 to 226.2 ± 4.0 ppm, respectively), before falling by 24 ppm to the 0–800 kyr observed glacial mean of 202.0 ± 3.2 ppm (Fig 3D)."*

**7. Line 421-422: There is an issue here with how the authors describe the antiphase hypothesis proposed by Raymo et al (2006). That publication did not propose an explanation for decreased obliquity sensitivity specifically, rather a change from local (insolation) forcing for a terrestrial ice margin to global (sea level/ice volume) forcing for a marine ice margin. Less total Antarctic ice volume change after the MPT resulted in less Antarctic ice volume sensitivity to all orbital cycles because Late Pleistocene Antarctic temperatures were consistently too cold for significant ice melt in East Antarctic. Therefore, I recommend changing this sentence to "is then proposed to remove sensitivity of Antarctic ice volume to local precession forcing in favor of quasi-100 kyr ice volume changes that are in phase between the hemispheres (Raymo et al., 2006)."**

Accepted and revised @ line 427-430:

> *"A transition from a smaller and more dynamic terrestrial-terminating Antarctic ice sheet to a larger and more stable marine-terminating ice sheet with cooling climate across the MPT (e.g. Elderfield et al., 2012) is then proposed to remove sensitivity of Antarctic ice volume to local precession forcing in favour of quasi-100 kyr ice volume changes that are in phase between the hemispheres (Raymo et al., 2006)."*

**8. Line 56: Change comma to semicolon "forcing; therefore, the mechanisms"**

Accepted and revised @ line 53:

**9. Line 58: Omit comma "A common element in many of these is internal…"**

Accepted and revised @ line 55:

**10. Line 82: Omit "to" so that the text says "changes fall into phase"**

Accepted and revised @ line 79:

**11. Line 116: "52" should instead be "57 globally-distributed records…" in the LR04 stack**

Accepted and revised @ line 113:

**12. Line 204: A negative sign appears to be missing in front of the slope of 33.37 because CO2 is lower when d18O is high (as shown in Fig. 2).**

Thanks for catching this. Accepted and revised @ line 199:

**13. Figure 3: The 95% confidence interval shading is very hard to see. I recommend making both the line color and shading color darker.**

Accepted and revised @ line 274 – Figure 3

**14. Line 338: I think "date" should be "data"**

Accepted and revised @ line 345

**15. Line 356: Remove extra "and" at the end of the sentence.**

Accepted and revised @ line 356:

**16. Line 402: Change "That fact that…" to "The fact that…"**

Accepted and revised @ line 410:

**17. Line 404: Insert "the" to say "that the LR04 benthic stack"**

Accepted and revised @ line 412:

**Correction of rounding errors and typos following review of code and data provision**

We uncovered, two minor rounding errors, one typo manuscript. Each error was in the decimal range, i.e. 0.1 ppm. and does not affect significance of any comparisons or conclusions. Below we present a table outlining the locations in our manuscript of these small changes. Apologies for these!

**Table R2:** Summary of minor changes to concentrations and uncertainties following review of code.

| Line | Data | Stage | i.e. | Period | Original | New | Reason |
|------|------|-------|------|--------|----------|-----|--------|
| 238 | Predictions | Undefined | Fig 3c | 0-800 ky | 225.1 | 225.2 | Rounding |
| 240 | Predictions | Undefined | Fig 3c | 1 Mya ± 89 kyr | 235.5 | 235.3 | Typo |
| 250 | Predictions | Glacial | Fig 3d | 0-800 ky | ± 1.7 | ± 1.6 | Rounding |

Data will be provided at the AAD data centre upon publication:
https://data.aad.gov.au/metadata/AAS_4632_Martin_etal_CP_2024